# From Parameters to Data: A Task-Parameter-Guided Fine-Tuning Pipeline for Efficient LLM Alignment

**Hao Chen**[1]  **Qi Zhang**[1]  **Liyao Li**[1]  **Zhanming Shen**[1]  **Wentao Ye**[1]  **Lirong Gao**[1]  **Ningtao Wang**[2]  **Xing Fu**[2]
**Xiaoyu Shen**[3]  **Junbo Zhao**[1]

## Abstract

Adapting Large Language Models (LLMs) to specialized domains typically incurs high data and computational overhead. While prior efficiency efforts have largely treated data selection and parameter-efficient fine-tuning as isolated processes, our empirical analysis suggests they may be intrinsically coupled. We posit the **Strong Map Hypothesis**: a sparse subset of attention heads plays a dominant role in task-specific adaptation, acting as keys that unlock specific data patterns. Building on this observation, we propose *From Parameters to Data (P2D)*, a unified framework that leverages these task-sensitive attention heads as a dual compass for both sample mining and structural pruning. To rigorously quantify the total pipeline cost, we introduce the Alignment Efficiency Ratio (AER) metric for both selection latency and training time. Mechanistically, **P2D** identifies critical heads via a lightweight proxy and uses them as a functional filter to curate high-affinity data, establishing a synergistic pipeline. Empirically, by updating merely 10% of attention heads on 10% of the data, **P2D** achieves an 8.3 pp performance gain over strong baselines and delivers a 7.0× end-to-end time speedup. These results validate that precise parameter-data synchronization eliminates redundancy, offering a new paradigm for efficient alignment.

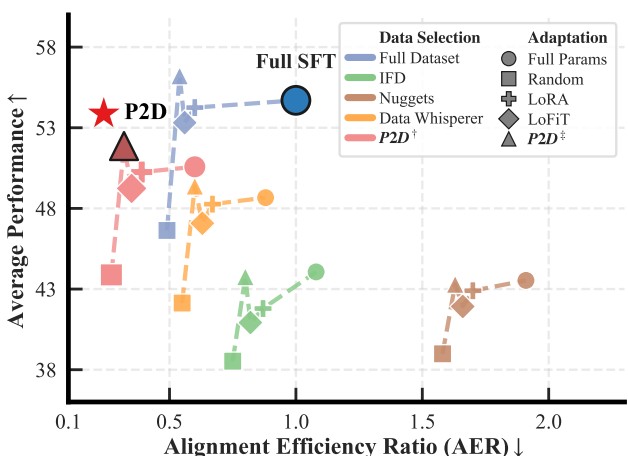

*Figure 1.* Comparison of AER↓ and performance↑. **P2D** (marked by ⋆) achieves the optimal trade-off, outperforming other strong baselines. The dashed lines connect adaptation variants for each selection strategy. Notably, P2D synergizes parameter-guided data selection (P2D[†]) with sparse head adaptation (P2D[‡]) for superior efficiency. Full SFT utilizes all data and parameters.

## 1. Introduction

With the tremendous momentum of large language models (LLMs) (Achiam et al., 2023; Guo et al., 2025; Dubey et al., 2024; Yang et al., 2025a), leveraging foundation models

for downstream applications has become central. While In-Context Learning (ICL) offers parameter-free adaptation, it often struggles in specialized domains requiring rigorous reliability, making fine-tuning the primary strategy for unlocking model potential (Mosbach et al., 2023; Liu et al., 2022). However, aligning general-purpose models is resource-intensive, involving massive data curation and infrastructure costs (Shao et al., 2024; Yang et al., 2024). This raises a central question: ***how can a general-purpose LLM be efficiently aligned to a downstream task?*** Consequently, there is an urgent need for alignment paradigms that substantially reduce data and compute overheads without compromising task performance (Zhao et al., 2023; Wan et al., 2023).

To improve efficiency, prior research has largely pursued two orthogonal directions. Data selection focuses on identifying high-quality subsets to match full-dataset performance with fewer samples (Li et al., 2024a; Wang et al., 2025). Conversely, parameter-efficient fine-tuning (PEFT) reduces adaptation costs by freezing the backbone and updating only a small fraction of parameters (Hu et al., 2022; Chen et al.,

[1]Zhejiang University [2]Ant Group [3]Eastern Institute of Technology. Correspondence to: Hao Chen <h.c.chen@zju.edu.cn>, Junbo Zhao <j.zhao@zju.edu.cn>.

2025). Crucially, *treating these processes in isolation overlooks their intrinsic coupling*. A data selection strategy optimized for full fine-tuning may be suboptimal for sparse parameter configurations. *We argue that data selection and fine-tuning are not independent levers but mutually reinforcing*: task-relevant signals reside both in the data and within the model, and the model itself can serve as a functional hook to guide the discovery of downstream-useful examples (Xia et al., 2024; Qin et al., 2024; Humane et al., 2025). Integrating these perspectives enables a cohesive framework where the selected data and the fine-tuning strategy are jointly optimized, forming a synergistic pipeline that surpasses disjoint pipeline combinations.

This perspective is grounded in a critical scientific observation regarding the interplay between model structures and data signals. Through extensive analysis, we posit the **Strong Map Hypothesis**: *a sparse subset of attention heads consistently plays a dominant role in task-specific adaptation, acting as implicit "keys" that unlock specific data patterns.* Our experiments suggest that alignment efficiency is not governed strictly by scale, but by the precision of this correspondence. This observation motivates a shift from dense to sparse structural alignment, revealing a latent opportunity: by pinpointing these critical parameter-data pairs, we can achieve substantial performance gains with a vanishingly small fraction of resources.

Building on this hypothesis, we propose *From Parameters to Data (P2D)*, a unified pipeline that exploits the model's intrinsic task response as a dual compass. Specifically, **P2D** unfolds in three stages: **i) Fast Head Identification** locates task-sensitive attention heads via a lightweight proxy; **ii) Parameter-Guided Data Selection** filters for samples that explicitly activate these functional components; and **iii) Sparse Head Adaptation** fine-tunes only these critical heads. To rigorously validate this unified paradigm and address the limitation of existing metrics that ignore selection overhead (Wang et al., 2025; Li et al., 2024a;b; Chen et al., 2024), we further introduce the **Alignment Efficiency Ratio (AER)**, a holistic metric that normalizes the total alignment cost (curation and adaptation) against full fine-tuning. Empirically, **P2D** updates merely 10% of attention heads using 10% of data yet achieves an 8.3 pp improvement and a 7.0× speedup, proving that we can unlock substantial performance by pinpointing these critical parameter-data pairs. This finding highlights a pivotal future direction: *decoding the intrinsic structural resonance between model and data signals for synergistic pipeline adaptation.*

In summary, our contributions are as follows:

- We posit the existence of a **Strong Map Hypothesis**: a sparse subset of attention heads consistently dominates task adaptation. This observation motivates a shift from dense to sparse structural alignment.

- We propose **P2D**, a unified framework that utilizes **task-specific attention heads as a functional hook** to drive data mining and structural parameter pruning jointly.
- We introduce the **Alignment Efficiency Ratio (AER)** to quantify end-to-end costs, and extensive experiments show **P2D** yields a 7.0× speedup over computational-heavy baselines with 8.3 pp performance improvement.

## 2. Related Work

### 2.1. Model-Centric Efficient Fine-Tuning

Model-centric approaches typically improve alignment efficiency through two primary paradigms: additive adaptation and selective utilization. **Additive methods**, broadly categorized as Parameter-Efficient Fine-Tuning (PEFT), freeze the pre-trained backbone and update only a minimal set of auxiliary parameters. Representative techniques include Low-Rank Adaptation (LoRA) (Hu et al., 2022), which injects trainable low-rank matrices, and adapter-based methods (Houlsby et al., 2019; Liu et al., 2024a) that insert lightweight modules between transformer layers. Other variants optimize continuous soft prompts (Zhao et al., 2024) to encode task-specific knowledge. Conversely, **selective methods** fine-tune only critical internal components, grounded in findings that specific heads dominate downstream tasks (Zhou et al., 2025; Shi et al., 2024). Building on this, methods like ALPS (Chen et al., 2025), and LOFiT (Yin et al., 2024) propose locating and fine-tuning only task-relevant sub-modules or representations while freezing the rest. While these approaches significantly reduce computational overhead, they predominantly treat parameter identification solely as a pruning tool. Our method repurposes these identified modules as a guidance signal for data selection, establishing a synergistic pipeline between model structure and training data.

### 2.2. Data-Centric Efficient Data Selection

Data-centric methods enhance alignment efficiency by filtering redundancy to construct high-quality subsets (Chen et al., 2024). Existing approaches predominantly focus on two dimensions: metric-based quality estimation and distributional diversity. **Metric-based methods** quantify sample utility via direct model feedback. For instance, IFD (Li et al., 2024a) and Nuggets (Li et al., 2024b) estimate the difficulty of each instance based on loss or perplexity variance, prioritizing samples that the model finds most informative. **Diversity-based methods**, conversely, aim to maximize information coverage. Methods like Data Whisperer (Wang et al., 2025) and Recost (Zhang et al., 2024) leverage embedding clustering or gradient matching to capture diverse semantic features (Liu et al., 2024b). Crucially, most prior works decouple the selection metric from the adaptation method, often relying on global statistics or external proxies.

We bridge this gap by strictly aligning selection with the model's intrinsic structure, choosing data that specifically resonates with the sparse attention heads targeted for update.

# 3. Method

## 3.1. Preliminary

**Problem Formulation.** We consider the scenario of aligning a pre-trained Large Language Model (LLM), parameterized by $\theta$, to a specific downstream task. The inputs consist of the pre-trained model $\mathcal{M}_\theta$ and a comprehensive labeled task dataset $\mathcal{D} = \{(x_i, y_i)\}_{i=1}^N$, where $x_i$ denotes the instruction/input and $y_i$ the target response. The standard alignment approach, Full Fine-Tuning (FFT), updates all parameters $\theta$ on the entire dataset $\mathcal{D}$ via supervised learning, incurring a time cost $t_{\text{FFT}}$.

Our goal is to construct an efficient alignment pipeline $f = (f_{ds}, f_{ft})$ that minimizes the total wall-clock time while maintaining task performance. This involves two sub-problems: (1) **Data Selection:** $f_{ds}$ selects a representative subset $\mathcal{D}_\mathcal{T} \subset \mathcal{D}$ with ratio $\rho_D = |\mathcal{D}_\mathcal{T}|/|\mathcal{D}|$; (2) **Efficient Head Adaptation:** $f_{ft}$ updates only a specific subset $\mathcal{H}_\mathcal{T}$ of full attention heads $\mathcal{H}$ ($\mathcal{H}_\mathcal{T} \subset \mathcal{H} \subset \theta$) with ratio $\rho_P = |\mathcal{H}_\mathcal{T}|/|\mathcal{H}|$ on $\mathcal{D}_\mathcal{T}$. The fine-tuning follows the standard instruction-tuning format, maximizing the conditional probability $p(y|x)$ without relying on few-shot examples in the context during inference.

To rigorously evaluate the efficiency gain, we introduce the **Alignment Efficiency Ratio (AER)**:

$$
\begin{aligned}
\text{AER}(f) &= \frac{t_f}{t_{FFT}}, \\
t_f &= \underbrace{t(f_{ds}, \rho_D)}_{\text{data-selection time}} + \underbrace{t(f_{ft}, \rho_P, \mathcal{D}_\mathcal{T})}_{\text{adaptation time}},
\end{aligned} \tag{1}
$$

where $t(f_{ds}, \rho_D)$ includes all pre-processing latencies (e.g., proxy model training, scoring), and $t(f_{ft}, \dots)$ represents the adaptation time. An AER $< 1$ indicates effective end-to-end acceleration, ensuring that pre-processing overheads do not negate training gains.

**Task-Specific Attention Heads Identification.** Building on mechanistic interpretability (Voita et al., 2019; Zhao et al., 2024), we posit that task-specific capabilities in Transformer models are localized within specific attention heads. Consider a model with $n$ heads. Each head $h$ computes an output $\boldsymbol{O}^h \in \mathbb{R}^{t \times d_v}$ via projection matrices $\{\boldsymbol{W}_q^h, \boldsymbol{W}_k^h, \boldsymbol{W}_v^h\}$. We define **task-specific heads** $\Theta_\mathcal{T}$ as those whose removal causes the most significant degradation on a task $\mathcal{T}$ with evaluation metric $p$:

$$
\Delta p(\theta^h) = p\left(\Theta^\mathcal{M}; \mathcal{T}\right) - p\left(\Theta^\mathcal{M} \setminus \theta^h; \mathcal{T}\right), \tag{2}
$$

where $\Theta^\mathcal{M}$ stands for the entire model parameters, and $\theta^\mathcal{M} \setminus \theta^h$ denotes the model without head parameters $\theta^h$. By ranking all $n$ heads by $\Delta p(\theta^h)$ and selecting the top-interested parameters top-$\rho_P$, we assemble a set of attention heads:

$$
\Theta_{\mathcal{T}, \rho_P} = \text{Top-}\rho_P\{\theta^h : \Delta p(\theta^h)\}, \tag{3}
$$

whose removal incurs the steepest drop on $\mathcal{T}$, thereby spotlighting the heads most critical for task performance.

**Task-Specific Data Selection.** Similarly, we define *task-specific data* as the subset of training examples that contribute most to the adaptation. Given the dataset $\mathcal{D}$, we aim to identify a subset $\mathcal{D}_\mathcal{T}$ of size $\rho_D|\mathcal{D}|$ such that the model trained on $\mathcal{D}_\mathcal{T}$ performs comparably to one trained on $\mathcal{D}$. This implies selecting samples $(x_i, y_i)$ with high influence scores $\Delta p(x_i, y_i)$, indicating their critical role in learning the task distribution:

$$
\begin{aligned}
\Delta p(x_i, y_i) = \, &p\left(\mathcal{M}(\mathcal{D}); \mathcal{T}\right) \\
&- p\left(\mathcal{M}(\mathcal{D} \setminus (x_i, y_i)); \mathcal{T}\right),
\end{aligned} \tag{4}
$$

where $\mathcal{M}(\mathcal{D})$ stands for a full-dataset fine-tuned model and $\mathcal{M}(\mathcal{D} \setminus (x_i, y_i))$ represents a model fine-tuned on the dataset without $(x_i, y_i)$. The interested subset $\mathcal{D}_\mathcal{T}$ should ensure that the performance of the fine-tuned model $\mathcal{M}(\mathcal{D}_\mathcal{T})$ remains comparable to the full dataset fine-tuned model $\mathcal{M}(\mathcal{D})$, and leads a subset dataset:

$$
\mathcal{D}_\mathcal{T} = \mathcal{D}_{\mathcal{T}, \rho_D} = \text{Top-}\rho_D\{(x_i, y_i) : \Delta p(x_i, y_i)\}, \tag{5}
$$

whose removal incurs a significant performance drop on $\mathcal{T}$.

## 3.2. From Parameters to Data: P2D

To fully exploit the intrinsic functionality and inductive biases of LLMs, and to synergistically enhance alignment efficiency across both data and parameter dimensions, we propose the *Parameters to Data (P2D)* framework (Fig. 2). P2D integrates three stages: (i) **Fast Head Identification**, which *rapidly* generates a lightweight proxy model to pinpoint task-sensitive attention heads; (ii) **Parameter-Guided Data Selection** (P2D†), which leverages these heads as "neural probes" to curate high-quality training examples; and (iii) **Sparse Head Adaptation** (P2D‡), which efficiently updates only the parameters of the identified task-sensitive heads using the curated subset.

### 3.2.1. STAGE I: FAST HEAD IDENTIFICATION

We identify task-specific attention heads by measuring the distributional shift in their induced transformations.

**Lightweight Proxy Construction.** Crucially, to avoid the high cost of full fine-tuning, we first construct a lightweight proxy model $\mathcal{M}_\mathcal{T}$. This proxy is obtained by fine-tuning

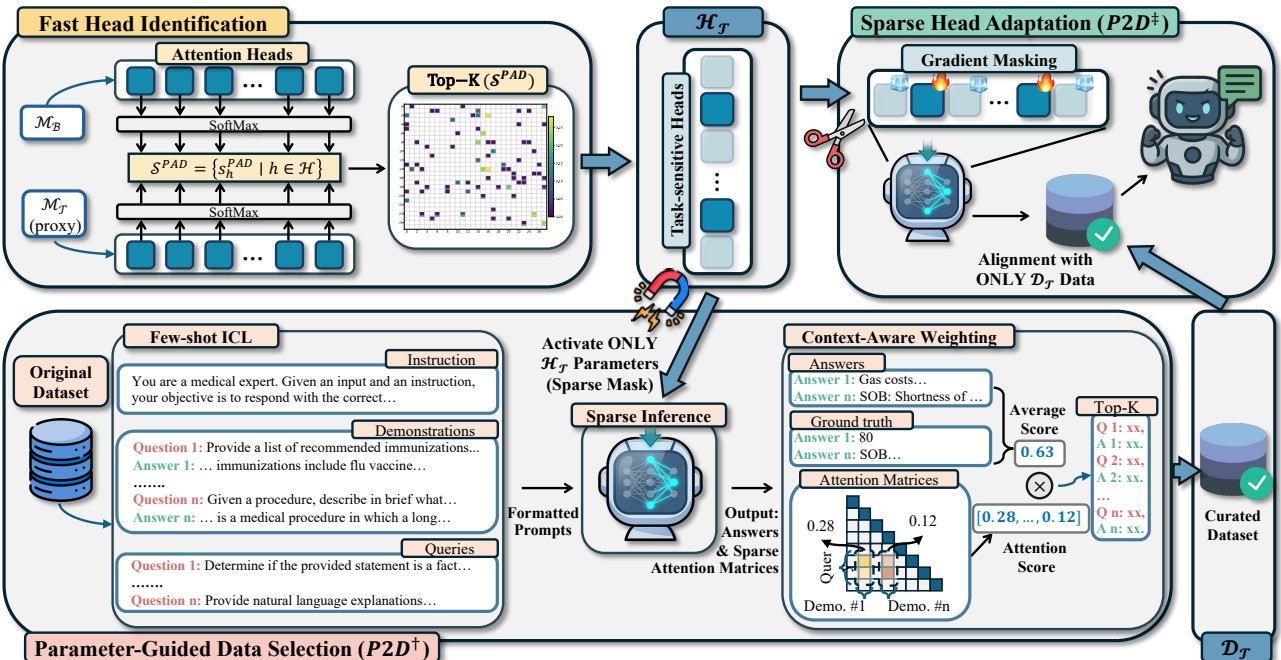

*Figure 2.* The overall framework of **P2D**, comprising three integral stages: **i) Fast Head Localization**, which identifies task-sensitive attention heads (denoted as $\mathcal{H}_T$ via a lightweight proxy; **ii) Parameter-Guided Data Selection (P2D$^\dagger$)**, which utilizes $\mathcal{H}_T$ as a sparse mask during inference to compute attention-based scores for curating a task-specific dataset $\mathcal{D}_T$; and **iii) Sparse Head Adaptation (P2D$^\ddagger$)**, which selectively updates only the parameters corresponding to $\mathcal{H}_T$ using the curated data $\mathcal{D}_T$. $\mathcal{M}_B$ and $\mathcal{M}_T$ denote the base model and the task-specific (or proxy) model, respectively. Details are illustrated in Section 3.2.

the base model $\mathcal{M}_B$ for a negligible number of steps (20 steps) on a tiny, randomly sampled subset of data (100 examples) using a full-batch update strategy. This process incurs minimal computational overhead, which is rigorously factored into the $t(f_{ds})$ term of our AER metric.

**Sensitivity Scoring.** To efficiently quantify the functional shift of each head without incurring inference costs, we analyze the static parameter space. For each head $h$, we define a composite proxy matrix $\boldsymbol{W}_{comp}^h$ that encapsulates the head's potential transformation capacity independent of input data $\boldsymbol{W}_{comp}^h = \boldsymbol{W}_q^h \boldsymbol{W}_k^{h\top} \boldsymbol{W}_v^h$, where $\boldsymbol{W}_q^h, \boldsymbol{W}_k^h, \boldsymbol{W}_v^h \in \mathbb{R}^{d_{model} \times d_{head}}$ are the projection weights. This composition effectively collapses the query-key interaction topology and the value projection into a single linear mapping, serving as a holistic signature of the head's mechanics. To measure the distributional shift, we flatten $W_{comp}^h$ into a vector and convert it into a probability distribution $\boldsymbol{P}^h$ via a global Softmax $P^h = \text{Softmax}(\text{Flatten}(\boldsymbol{W}_{comp}^h)/\tau)$. The task sensitivity score $s_h^{PAD}$ is then computed as the Wasserstein-1 distance between the distributions derived from the base model and the proxy model:

$$
\begin{aligned}
s_h^{PAD} &= W_1\left(\boldsymbol{P}_{\mathcal{B}}^h, \boldsymbol{P}_{\mathcal{T}}^h\right) \\
&= \inf_{\gamma \in \Gamma(\boldsymbol{P}_{\mathcal{B}}^h, \boldsymbol{P}_{\mathcal{T}}^h)} \mathbb{E}_{(x,y)\sim\gamma}\left[\|x - y\|\right],
\end{aligned} \quad (6)
$$

where $\Gamma(\boldsymbol{P}_{\mathcal{B}}^h, \boldsymbol{P}_{\mathcal{T}}^h)$ is the set of all couplings with marginals $\boldsymbol{P}_{\mathcal{B}}^h$ and $\boldsymbol{P}_{\mathcal{T}}^h$. We adopt $S^{PAD}$ as it has been empirically verified to outperform magnitude-based metrics (Chen et al., 2025). Computing $s_h^{PAD}$ for all heads $h \in \mathcal{H} = \{1, \dots, n\}$ yields the collection $\mathcal{S}^{PAD} = \{s_h^{PAD} \mid h \in \mathcal{H}\}$. We then define the task-sensitive head set $\mathcal{H}_{\mathcal{T}}$ as the subset of heads corresponding to the top $\rho_P$ fraction of scores in $\mathcal{S}^{PAD}$, i.e., those with the largest $s_h^{PAD}$ values. We choose $W_1$ over activation-gradient and cosine-similarity alternatives for its linear sensitivity to small parameter drifts and its data-free, near-zero scoring cost (Appendix E).

### 3.2.2. STAGE II: PARAMETER-GUIDED DATA SELECTION

Given the set of task-sensitive heads $\mathcal{H}_{\mathcal{T}}$ identified in the first stage, we aim to select data that explicitly activates these functional components. While prior work like Data Whisperer (Wang et al., 2025) aggregates attention signals globally across the entire model, we argue that this introduces noise from task-irrelevant modules. Instead, **P2D** enforces a strictly **functional alignment**: utilizing the identified heads as a *denoising filter* to rank training examples based on their contribution to the specific downstream task.

**ICL-Based Probing.** We first evaluate candidate examples via their influence on In-Context Learning (ICL)

performance. Let $\mathcal{D}$ be the candidate pool. In each iteration, we sample a demonstration set $\mathcal{D}_d = \{(x_d^{(1)}, y_d^{(1)}), \ldots, (x_d^{(n_d)}, y_d^{(n_d)})\}$ and a disjoint query set $\mathcal{D}_q$. We construct a context $C$ with a task instruction $I$:

$$C = \{I, (x_d^{(1)}, y_d^{(1)}), \ldots, (x_d^{(n_d)}, y_d^{(n_d)})\}. \quad (7)$$

The model $\mathcal{M}$ predicts the query targets conditioned on $C$. We calculate a base performance score $s$ for the examples in $\mathcal{D}_d$ based on the prediction accuracy on $\mathcal{D}_q$:

$$s = \frac{1}{n_q} \sum_{j=1}^{n_q} f(\hat{y}_q^{(j)}, y_q^{(j)}), \quad (8)$$

where $f(\cdot)$ is the evaluation metric (average accuracy or ROUGE-L (Lin, 2004)).

**Structure-Aware Filtering.** To incorporate structural signals, we refine the scores using the attention patterns of the *identified* task-sensitive heads. Let $A^{(h)}$ be the self-attention matrix for head $h$. For each demonstration $(x_d^{(i)}, y_d^{(i)}) \in \mathcal{D}_d$, let $A_{(x_d^{(i)}, y_d^{(i)})}^{(h)}$ denote the sub-matrix capturing the attention weights from the query tokens to this specific demonstration.

Crucially, unlike global aggregation methods, we compute the importance weight by accumulating attention **only within the task-sensitive heads** $\mathcal{H}_{\mathcal{T}}$:

$$w_{(x_d^{(i)}, y_d^{(i)})} = \frac{1}{\ell(x_d^{(i)})} \sum_{h \in \mathcal{H}_{\mathcal{T}}} \mathbf{1}^{\top} A_{(x_d^{(i)}, y_d^{(i)})}^{(h)} \mathbf{1}. \quad (9)$$

Here, $\ell(x_d^{(i)})$ is the token length for normalization. By restricting the summation to $h \in \mathcal{H}_{\mathcal{T}}$, this weight $w$ specifically measures the example's utility in activating the task-critical parameters, effectively filtering.

**Iterative Scoring and Selection.** We perform $T$ iterations of the ICL probing. In each iteration $t$, a demonstration set $\mathcal{D}_d$ and a query set $\mathcal{D}_q$ are sampled, yielding a base task score $s^{(t)}$ (as in Eq. 8). For each specific example $x_i \in \mathcal{D}_d$ involved in this iteration, we compute its structural weight $w_i^{(t)}$ (Eq. 9) and accumulate its importance contribution (specifically, $v_i^{(t)} = s^{(t)} \cdot w_i^{(t)}$). After $T$ iterations, the final importance score for sample $x_i$ is computed as the average over its selection frequency $n_i$:

$$s_{final}(x_i) = \frac{1}{n_i} \sum_{k=1}^{n_i} v_i^{(k)} = \frac{1}{n_i} \sum_{k=1}^{n_i} \left( s^{(k)} \cdot w_i^{(k)} \right). \quad (10)$$

Finally, we rank all candidates by $s_{final}$ and select the top-$\rho_D$ subset to form the task-specific dataset $\mathcal{D}_{\mathcal{T}}$.

### 3.2.3. STAGE III: SPARSE HEAD ADAPTATION

In the final stage, we perform parameter-efficient fine-tuning using the curated subset $\mathcal{D}_{\mathcal{T}}$ and the identified heads $\mathcal{H}_{\mathcal{T}}$.

**Gradient Masking.** We freeze all parameters in the model except for the projection matrices $\{\theta^h\}_{h \in \mathcal{H}_{\mathcal{T}}}$ of the task-sensitive heads. Let $m_h = \mathbb{I}[h \in \mathcal{H}_{\mathcal{T}}]$ be the indicator for whether head $h$ is task-relevant. The gradient updates are effectively masked as:

$$\nabla_{\theta^h} \mathcal{L} \leftarrow m_h \cdot \nabla_{\theta^h} \mathcal{L}, \quad (11)$$

ensuring that only heads in $\mathcal{H}_{\mathcal{T}}$ receive non-zero updates. Equivalently, the optimizer is applied solely to $\{\theta^h\}_{h \in \mathcal{H}_{\mathcal{T}}}$, while parameters for $h \notin \mathcal{H}_{\mathcal{T}}$ remain frozen.

**Optimization Objective.** The fine-tuning objective over the selected data is the expected negative log-likelihood:

$$\mathcal{L} = \mathbb{E}_{(x,y) \sim \mathcal{D}_{\mathcal{T}}} \left[ -\log p\left(y \mid x; \{\theta^h\}_{h \in \mathcal{H}_{\mathcal{T}}}\right) \right]. \quad (12)$$

This targeted update reduces optimization redundancy by concentrating capacity on the heads most sensitive to the downstream task, thereby preserving the broader pretrained knowledge encoded in the frozen heads. The full procedure is summarized in Appendix B.

## 4. Experiments

### 4.1. Experimental Setup

**Datasets and Models.** We evaluate on three diverse datasets: BioInstruct (Tran et al., 2024) (biomedical), DialogSum (Chen et al., 2021) (dialogue), and GSM8K (Cobbe et al., 2021) (math reasoning). For models, we employ Qwen-2.5-7B-Instruct (Yang et al., 2025b), Qwen-3-8B (Yang et al., 2025a), and Llama-3-8B-Instruct (Dubey et al., 2024). Details in Appendix A.1.

**Baselines.** We compare **P2D** against two categories of baselines: **1) Data Selection:** Beyond training on the Full Dataset, we include IFD (Li et al., 2024a) (instruction-following difficulty), Nuggets (Li et al., 2024b) (one-shot scoring), and Data Whisperer (Wang et al., 2025) (ICL-based selection). **2) Adaptation:** We compare against Full Parameter fine-tuning, LoRA (Hu et al., 2022), Random which updates randomly selected attention heads, and LoFiT (Yin et al., 2024) which identifies and updates task-relevant representations. For LoRA, we report $r=128$ in the main tables (Appendix G). All results are averaged across three random seeds. More details are in Appendix A.2, J.

**Implementation.** We report Exact Match (EM) for GSM8K and ROUGE-L (Lin, 2004) for generation tasks. All experiments are conducted on eight NVIDIA A100 GPUs. See Appendix A for hyperparameter details.

### 4.2. Results

**Main Results.** Table 1 reports the performance across GSM8K, DialogSum, and BioInstruct under a strict budget

*Table 1.* Evaluation of various data selection and fine-tuning strategies on GSM8K, DialogSum, and BioInstruct using Qwen-2.5-7B-Instruct (Q2.5-7), Qwen-3-8B (Q3-8), and Llama-3-8B-Instruct (L3-8). We report the performance of varying the Data Ratio ($\rho_D$) and Head Ratio ($\rho_P$). The gray-colored row represents the Full SFT baseline, while the subsequent blocks use specific data selection methods constrained to 10% of the data. Within each block, we compare our proposed strategy with other fine-tuning baselines. P2D$^\dagger$ denotes our proposed Parameter-Guided Data Selection, while P2D$^\ddagger$ represents our Sparse Head Adaptation. Best results are highlighted in **bold**.

| Method | $\rho_D$ | $\rho_P$ | GSM8K | | | DialogSum | | | BioInstruct | | | Avg. |
|---|---|---|---|---|---|---|---|---|---|---|---|---|
| | | | Q2.5-7 | Q3-8 | L3-8 | Q2.5-7 | Q3-8 | L3-8 | Q2.5-7 | Q3-8 | L3-8 | |
| *Vanilla* | – | – | *62.31* | *65.73* | *48.62* | *25.41* | *22.97* | *20.16* | *16.35* | *12.94* | *12.92* | *31.93* |
| *Full (full dataset)* | | | | | | | | | | | | |
| + full parameters | 100% | 100% | 81.55 | 84.89 | 68.63 | 48.36 | 44.22 | 46.89 | 38.01 | **42.16** | 37.59 | 54.70 |
| + random | 100% | 10% | 68.12 | 70.45 | 58.33 | 41.56 | 39.23 | 40.77 | 32.84 | 38.47 | 29.95 | 46.64 |
| + LoRA | 100% | 10% | 80.14 | 82.56 | 67.32 | 49.18 | 44.09 | 46.72 | 38.45 | 41.68 | 38.12 | 54.25 |
| + LoFiT | 100% | 10% | 79.23 | 81.45 | 66.12 | 48.34 | 43.11 | 45.89 | 37.56 | 40.89 | 37.23 | 53.31 |
| + *P2D$^\ddagger$ (Ours)* | 100% | 10% | **83.03** | **86.12** | **69.51** | **50.67** | **45.89** | **48.15** | **41.92** | 41.40 | **39.28** | **56.22** |
| *IFD* | | | | | | | | | | | | |
| + full parameters | 10% | 100% | 75.12 | 77.04 | 56.45 | 37.10 | **35.22** | 34.07 | **26.78** | **28.56** | **26.23** | **44.06** |
| + random | 10% | 10% | 66.78 | 66.23 | 52.65 | 32.87 | 30.98 | 29.45 | 23.29 | 22.85 | 21.66 | 38.53 |
| + LoRA | 10% | 10% | 74.34 | 75.08 | 55.13 | 35.45 | 33.02 | 31.31 | 23.98 | 24.56 | 23.27 | 41.79 |
| + LoFiT | 10% | 10% | 73.45 | 74.12 | 54.23 | 34.56 | 32.11 | 30.45 | 23.12 | 23.78 | 22.45 | 40.92 |
| + *P2D$^\ddagger$ (Ours)* | 10% | 10% | **77.33** | **78.90** | **58.03** | **38.67** | 35.18 | **34.33** | 24.34 | 23.97 | 23.15 | 43.77 |
| *Nuggets* | | | | | | | | | | | | |
| + full parameters | 10% | 100% | 73.28 | **76.12** | 55.28 | **36.38** | 32.21 | 36.02 | 27.78 | 26.62 | **28.13** | 43.54 |
| + random | 10% | 10% | 65.11 | 67.89 | 52.65 | 31.85 | 29.98 | 32.98 | 22.92 | 23.99 | 23.54 | 38.99 |
| + LoRA | 10% | 10% | 72.34 | 74.56 | 53.12 | 34.45 | **33.78** | 35.93 | **28.45** | **27.34** | 26.12 | 42.90 |
| + LoFiT | 10% | 10% | 71.56 | 73.21 | 52.45 | 33.67 | 32.89 | 34.12 | 27.56 | 26.45 | 25.34 | 41.92 |
| + *P2D$^\ddagger$ (Ours)* | 10% | 10% | **74.32** | 75.32 | **57.08** | 35.43 | 33.01 | **36.78** | 27.52 | 25.31 | 24.89 | 43.30 |
| *Data Whisperer* | | | | | | | | | | | | |
| + full parameters | 10% | 100% | 79.05 | 81.23 | 63.45 | 43.67 | 41.34 | 39.56 | 29.01 | 31.23 | **29.45** | 48.67 |
| + random | 10% | 10% | 70.73 | 71.15 | 56.38 | 38.12 | 36.45 | 33.78 | 23.34 | 26.01 | 23.23 | 42.13 |
| + LoRA | 10% | 10% | 77.34 | 79.56 | 64.12 | 43.89 | **42.45** | **40.78** | 27.45 | 30.67 | 28.12 | 48.26 |
| + LoFiT | 10% | 10% | 76.12 | 78.34 | 62.89 | 42.56 | 41.21 | 39.45 | 26.56 | 29.34 | 27.23 | 47.08 |
| + *P2D$^\ddagger$ (Ours)* | 10% | 10% | **80.23** | **81.41** | **65.89** | **44.12** | 42.01 | 40.23 | **30.34** | **31.56** | 28.67 | **49.38** |
| *P2D$^\dagger$ (Ours)* | | | | | | | | | | | | |
| + full parameters | 10% | 100% | 80.05 | 82.13 | 66.78 | 45.34 | 43.01 | 41.56 | 31.12 | 33.45 | 31.78 | 50.58 |
| + random | 10% | 10% | 72.89 | 70.95 | 57.56 | 39.89 | 38.45 | 37.97 | 26.56 | 24.78 | 25.89 | 43.88 |
| + LoRA | 10% | 10% | 79.45 | 80.67 | 67.34 | 45.01 | 42.12 | 41.45 | 30.89 | **34.34** | 31.01 | 50.25 |
| + LoFiT | 10% | 10% | 78.56 | 79.45 | 66.11 | 44.12 | 41.21 | 40.56 | 29.78 | 33.12 | 30.23 | 49.24 |
| + *P2D$^\ddagger$ (Ours)* | 10% | 10% | **82.56** | **84.78** | **69.12** | **46.45** | **43.23** | **42.45** | **32.78** | 32.89 | **32.56** | **51.87** |

(typically 10% data and attention heads). Our unified framework, **P2D** (the last row, coupling parameter-guided data selection **P2D$^\dagger$** with sparse head adaptation **P2D$^\ddagger$**), consistently achieves the best performance. On GSM8K, the full P2D pipeline not only outperforms other constrained baselines but also rivals the computational-heavy full-data training. Crucially, on **domain-specific tasks** like DialogSum and BioInstruct, vanilla models often lack intrinsic knowledge and thus typically rely heavily on extensive training data, and P2D effectively identifies the most information-dense examples. These results highlight three key findings: (1) **Superior Selection:** Our selection module P2D$^\dagger$ consistently surpasses existing baselines; (2) **Synergy:** Coupling selection with targeted fine-tuning (P2D$^\ddagger$) yields significant gains over disjoint optimization; and (3) **Less is More:**

By focusing on task-essential parameters and data, P2D mitigates the noise inherent in training, achieving better alignment. More results are in Appendix L.

**Efficiency and Performance Trade-off.** Table 2 reports the Alignment Efficiency Ratio (AER, lower is better) and average performance (Perf.) using 10% of the data. P2D achieves the optimal trade-off, recording the lowest AERs (e.g., 0.32 on GSM8K) alongside superior performance across all tasks. While the data-selection variant P2D$^\dagger$ is effective, our full pipeline integrates sparse training to maximize efficiency further. Compared to strong baselines like Nuggets and DW, P2D not only yields substantial speedups (up to 8.63×) but also breaks the conventional efficiency–accuracy trade-off. More details in Appendix K.

*Table 2.* Comparison of Alignment Efficiency Ratio (AER) and averaged performance (Perf.) over three models. All data selection methods utilize 10% examples. Sparse training variants update only 10% of the attention heads. Speedup reports the runtime improvement of our method over the Nuggets baseline (w/ full params). Best results are highlighted in **bold**.

| Method | GSM8K | | DialogSum | | BioInstruct | |
|---|---|---|---|---|---|---|
| | AER↓ | Perf.↑ | AER↓ | Perf.↑ | AER↓ | Perf.↑ |
| *Full Dataset* | | | | | | |
| + full params | 1.00 | 78.36 | 1.00 | 46.49 | 1.00 | 39.25 |
| + random | **0.49** | 65.63 | **0.49** | 40.52 | **0.49** | 33.75 |
| + LoRA | 0.60 | 76.67 | 0.60 | 46.66 | 0.60 | 39.42 |
| + LoFiT | 0.56 | 75.60 | 0.56 | 45.78 | 0.56 | 38.56 |
| + P2D$^{\ddagger}$ | 0.54 | **79.55** | 0.54 | **48.24** | 0.54 | **40.87** |
| *IFD* | | | | | | |
| + full params | 1.08 | 69.54 | 1.18 | 35.46 | 1.12 | **27.19** |
| + random | **0.75** | 61.89 | **0.85** | 31.10 | **0.79** | 22.60 |
| + LoRA | 0.87 | 68.18 | 0.97 | 33.26 | 0.91 | 23.94 |
| + LoFiT | 0.82 | 67.27 | 0.92 | 32.38 | 0.86 | 23.12 |
| + P2D$^{\ddagger}$ | 0.80 | **71.42** | 0.90 | **36.06** | 0.84 | 23.82 |
| *Nuggets* | | | | | | |
| + full params | 1.91 | 68.23 | 3.28 | 34.87 | 1.62 | **27.51** |
| + random | **1.58** | 61.88 | **2.95** | 31.60 | **1.29** | 23.48 |
| + LoRA | 1.70 | 66.67 | 3.07 | 34.72 | 1.41 | 27.30 |
| + LoFiT | 1.65 | 65.75 | 3.02 | 33.56 | 1.36 | 26.45 |
| + P2D$^{\ddagger}$ | 1.63 | **68.91** | 3.00 | **35.07** | 1.34 | 25.91 |
| *Data Whisperer* | | | | | | |
| + full params | 0.88 | 74.58 | 1.12 | 41.52 | 0.77 | 29.90 |
| + random | **0.55** | 66.09 | **0.79** | 36.12 | **0.44** | 24.19 |
| + LoRA | 0.67 | 73.67 | 0.91 | **42.37** | 0.56 | 28.75 |
| + LoFiT | 0.62 | 72.45 | 0.86 | 41.08 | 0.51 | 27.71 |
| + P2D$^{\ddagger}$ | 0.60 | **75.84** | 0.84 | 42.12 | 0.49 | **30.19** |
| *P2D$^{\dagger}$ (Ours)* | | | | | | |
| + full params | 0.60 | 76.32 | 0.66 | 43.30 | 0.53 | 32.12 |
| + random | **0.27** | 67.13 | **0.33** | 38.77 | **0.20** | 25.74 |
| + LoRA | 0.39 | 75.82 | 0.45 | 42.86 | 0.32 | 32.08 |
| + LoFiT | 0.34 | 74.71 | 0.40 | 41.97 | 0.27 | 31.04 |
| + P2D$^{\ddagger}$ | 0.32 | **78.82** | 0.38 | **44.04** | 0.25 | **32.74** |
| **Speedup** | **5.97×** | – | **8.63×** | – | **6.48×** | – |

## 4.3. Ablation Study

Tables 3 and 4 investigate data ($\rho_D$) and heads ($\rho_P$) ratios. **Data Ratio:** While reasoning tasks (GSM8K) show high redundancy (robust at 10% data), domain tasks benefit more from scale. **Heads Ratio:** Peak results typically emerge at low ratios (10%–30%), confirming the parameter redundancy; sparse updates prevent overfitting better than denser ones. We adopt $\rho_P = 10\%$ as the optimal default.

## 5. Discussion and Analysis

**Fast Head Identification (FHI).** We prioritize leveraging existing task-specific models (Readily $\mathcal{M}$, e.g., Qwen2.5-Math) for head identification. However, such models are often unavailable for specialized domains. In such cases, we propose **Fast Head Identification (FHI)** as an efficient alternative to costly SFT. FHI exploits the finding that ***task-sensitive structural patterns emerge early in training*** (Chen et al., 2025; Zhou et al., 2025; Zhao et al., 2024; Yin et al.,

*Table 3.* Ablation of the data selection ratio $\rho_D$ with $\rho_P$ fixed at 10% for P2D. Best results are in **bold**.

| Dataset | $\rho_D$ | Q2.5-7 | Q3-8 | L3-8 |
|---|---|---|---|---|
| GSM8K | 100% | 83.03 | 86.12 | **69.51** |
| | 70% | **83.85** | **86.40** | 69.32 |
| | 50% | 81.68 | 83.95 | 68.20 |
| | 30% | 82.20 | 84.13 | 68.88 |
| | 10% | 82.56 | 84.78 | 69.12 |
| DialogSum | 100% | **50.67** | **45.89** | **48.15** |
| | 70% | 50.10 | 45.67 | 47.72 |
| | 50% | 49.02 | 43.20 | 45.05 |
| | 30% | 47.55 | 44.13 | 45.57 |
| | 10% | 46.45 | 43.23 | 42.45 |
| BioInstruct | 100% | **41.92** | **41.40** | **39.28** |
| | 70% | 40.81 | 40.22 | 38.33 |
| | 50% | 36.05 | 37.72 | 35.55 |
| | 30% | 37.20 | 36.33 | 35.23 |
| | 10% | 32.78 | 32.89 | 32.56 |

*Table 4.* Ablation on head ratio $\rho_P$ with $\rho_D$ fixed at 10% for P2D. **Bold** indicates the best per model.

| Model | $\rho_P$ | GSM8K | DialogSum | BioInstruct |
|---|---|---|---|---|
| Q2.5-7 | 100% | 80.05 | 45.34 | 31.12 |
| | 70% | 81.48 | 44.92 | 30.87 |
| | 50% | 79.82 | 45.76 | 31.54 |
| | 30% | 82.13 | 46.07 | 32.61 |
| | 10% | **82.56** | **46.45** | **32.78** |
| Q3-8 | 100% | 82.13 | 43.01 | 33.45 |
| | 70% | 82.81 | 41.52 | **34.02** |
| | 50% | 81.05 | 39.71 | 31.11 |
| | 30% | 83.50 | 42.02 | 33.87 |
| | 10% | **84.78** | **43.23** | 32.89 |
| L3-8 | 100% | 66.78 | 41.56 | 31.78 |
| | 70% | 65.89 | 42.02 | 30.21 |
| | 50% | 68.33 | 41.82 | 31.03 |
| | 30% | 67.67 | 42.19 | 32.13 |
| | 10% | **69.12** | **42.45** | **32.56** |

2024). As shown in Table 6, we compare different acquisition strategies. By performing toy training on 100 examples, FHI obtains a proxy model sufficient to pinpoint critical components with negligible cost. While this proxy lacks inference capability due to data scarcity, it captures the structural sensitivity required to initialize our pipeline. More ablations on the proxy model are provided in Appendix D.

**Scaling Across Model Sizes.** Table 5 extends the evaluation to a broader scale on Qwen-2.5-Instruct. To isolate the contribution of *sparse head adaptation* from data selection, we restrict the comparison here to the full-dataset setting and vary only the parameter-update strategy. **P2D**$^{\ddagger}$

*Table 5.* Scaling P2D[‡] across model sizes from 1.5B to 32B on Qwen-2.5-Instruct. To isolate sparse head adaptation from data selection, all rows here use the *full dataset* and vary only the parameter-update strategy: Full SFT (100% params), LoRA (10% trainable), and our **P2D**[‡] (10% attention heads). Full per-scale tables including data-selection variants are in Appendix F.

| Method | 1.5B | | | 3B | | | 7B | | | 14B | | | 32B | | |
|---|---|---|---|---|---|---|---|---|---|---|---|---|---|---|---|
| | GSM | Dial. | Bio. | GSM | Dial. | Bio. | GSM | Dial. | Bio. | GSM | Dial. | Bio. | GSM | Dial. | Bio. |
| Full SFT | 63.45 | 29.34 | 18.56 | 72.56 | 36.78 | 24.63 | 81.55 | 48.36 | 38.01 | 86.47 | 50.93 | 43.12 | 88.94 | 53.12 | 45.78 |
| LoRA | 62.18 | 30.07 | 17.93 | 71.83 | 37.51 | 23.87 | 80.14 | 49.18 | 38.45 | 85.36 | 52.14 | 42.56 | 87.73 | 54.38 | 45.29 |
| **P2D**[‡] **(Ours)** | **65.23** | **31.48** | **19.74** | **74.61** | **38.19** | **25.74** | **83.03** | **50.67** | **41.92** | **88.21** | **53.47** | **45.89** | **90.23** | **55.34** | **47.89** |

*Table 6.* Comparisons of different model acquisitions.

| Method | GSM8K | DialogSum | BioInstruct |
|---|---|---|---|
| | *Qwen-2.5-7B-Instruct* | | |
| SFT | 81.90 | 47.10 | 33.20 |
| Readily $\mathcal{M}$ | 81.23 | – | – |
| *FHI* | 82.56 | 46.45 | 32.78 |
| | *Qwen-3-8B* | | |
| SFT | 85.10 | 44.20 | 33.65 |
| Readily $\mathcal{M}$ | – | – | – |
| *FHI* | 84.78 | 43.23 | 32.89 |
| | *Llama-3-8B-Instruct* | | |
| SFT | 68.48 | 43.05 | 33.10 |
| Readily $\mathcal{M}$ | 67.92 | – | – |
| *FHI* | 69.12 | 42.45 | 32.56 |

*Table 7.* Comparison between the FHI Proxy and SFT. The Proxy model serves only as a structural probe for head identification.

| Model | Samples | Params | AER↓ | Performance↑ | | |
|---|---|---|---|---|---|---|
| | | | | GSM8K | Dial. | BioI. |
| Full SFT | Full | 100% | 1.00 | 78.36 | 46.49 | 39.25 |
| FHI Proxy | 100 | 100% | **0.01** | 18.45 | 12.30 | 10.15 |
| **P2D (Final)** | 10% | **10%** | 0.35 | **78.82** | 44.04 | 32.74 |

*Table 8.* Stability of P2D's two stochastic components across independent seeds (Qwen-2.5-7B, GSM8K). **Head Overlap**: percentage of common head indices in the top-10% (79 of 784) of the FHI proxy across two proxy seeds. **Dataset Overlap**: pairwise Jaccard of the selected 10% data examples across two ICL demonstration seeds.

| Seed Pair | Head Overlap (%) | Dataset Overlap (%) |
|---|---|---|
| 42 vs. 43 | 92.4 | 93.3 |
| 42 vs. 44 | 88.6 | 92.8 |
| 43 vs. 44 | 91.1 | 93.1 |
| **Average** | **90.7** | **93.1** |

used during ICL probing for data scoring. We verify that both the identified task-sensitive heads and the selected 10% data subset are stable across seeds. Running each component from three independent seeds ($\{42, 43, 44\}$), Table 8 reports ~91% pairwise overlap on the top-10% heads (79 of 784) and ~93% Jaccard overlap on the curated 10% data. The head-overlap stability aligns with mechanistic-interpretability findings that task-supporting attention circuits emerge as stable computational building blocks early in training (Chen et al., 2025; Olsson et al., 2022); the data-selection stability is by design, as P2D's iterative randomized probing resamples demos across queries to attenuate the fixed-demo bias known in ICL (Min et al., 2022). Crucially, these two stability sources are independent yet both deterministic, suggesting that the identified Strong Map is a robust structural attribute of the pretrained backbone rather than a sampling artifact. We further analyze the disentangled origin of P2D's 7.0× end-to-end speedup in Appendix I, specifically, the relative contributions of parameter sparsity (~ 1.72×) and data sparsity (~ 2.22×) on top of P2D's near-zero selection overhead.

**Sparse Adaptation Mitigates Catastrophic Forgetting.** A natural concern with freezing the MLP layers and 90% of attention heads is whether sparse adaptation impairs the model's general capabilities. We empirically address this by aligning Qwen-2.5-7B on UltraChat (Ding et al., 2023) (10% data) and evaluating zero-shot on two general-purpose benchmarks (MMLU (Hendrycks et al., 2020), ARC-Challenge (Clark et al., 2018)). Table 9 shows that Full SFT, despite training on the same 10% subset, causes

outperforms both **Full SFT and LoRA at every scale across all three benchmarks**, with the absolute gain over Full SFT widening further with model scales up, confirming that the **Strong Map** becomes more structurally concentrated as parameter capacity grows. This is intuitive: **larger models exhibit greater parameter redundancy**, allowing sparse head adaptation to deliver compounding gains in both performance and efficiency. Notably, the same monotonic trend holds across mathematical reasoning, dialogue, and biomedical tasks, confirming that the Strong Map is a generic property of large-model attention geometry rather than a benchmark-specific artifact. Full tables are in Appendix F.

**Robustness to Random Seeds.** P2D has two independent sources of stochasticity: (i) the 100-sample draw used to build the FHI proxy, and (ii) the in-context demonstrations

*Table 9.* Catastrophic-forgetting evaluation. Qwen-2.5-7B is aligned on UltraChat with 10% data and evaluated zero-shot on MMLU and ARC-Challenge. P2D substantially mitigates forgetting versus Full SFT and LoRA, supporting that sparse head adaptation preserves pre-trained capabilities.

| Method | MMLU ↑ | ARC-C ↑ |
|---|---|---|
| Vanilla (no alignment) | 74.23 | 88.14 |
| Full SFT (10% data) | 56.34 | 61.89 |
| LoRA (r=128, 10% data) | 63.12 | 71.56 |
| **P2D (10%/10%)** | **67.45** | **78.23** |

*Table 10.* Comparison of average sample lengths (Tokens and Words) between the Full Dataset and the subset selected by P2D.

| Dataset | Avg. Tokens / Sample | | Avg. Words / Sample | |
|---|---|---|---|---|
| | Full Dataset | P2D (10%) | Full Dataset | P2D (10%) |
| **GSM8K** | 157.50 | 159.20 | 96.75 | 98.10 |
| **DialogSum** | 227.03 | 231.45 | 153.85 | 156.20 |
| **BioInstruct** | 93.50 | 95.80 | 70.92 | 72.50 |

catastrophic forgetting: MMLU drops from 74.23 to 56.34 and ARC-C from 88.14 to 61.89. LoRA partially mitigates this drop. In contrast, P2D's structural sparsity protects the pre-trained knowledge encoded in the frozen MLP layers and remaining heads, retaining substantially more general capability (67.45 MMLU, 78.23 ARC-C). This corroborates the recent academic shift toward attention-centric alignment (Chen et al., 2025; Yin et al., 2024): routing-level updates are sufficient to inject task-specific signal without overwriting the broad knowledge stored in MLP layers.

**Visualizing the Strong Map Hypothesis: Structure Side**
Figure 3 visualizes the top 10% task-sensitive attention heads identified by P2D on Qwen-2.5-7B. We observe a clear phenomenon of **Structural Specialization**: while some heads are broadly active (indicating general-purpose utility), distinct sparse clusters activate exclusively for specific tasks (e.g., GSM8K vs. BioInstruct). These heads span across multiple layers, forming a task-specific functional sub-network. **This empirically validates the structural side of our Strong Map Hypothesis**: downstream capabilities are not uniformly distributed but are intrinsically mapped to sparse, task-specific parameter groups. This explains why updating this targeted subset is more effective than broad parameter adaptation.

**Visualizing the Strong Map Hypothesis: Data Side** As illustrated in Figure 4, the subset curated by P2D exhibits consistently lower perplexity (PPL). A common concern is that low-PPL selection might trivially favor "easy" or "short" samples (e.g., simple greetings or short sentences) rather than task-relevant content. To investigate this, we conducted a quantitative analysis comparing the average length of samples in the full dataset versus the P2D-selected subset. Table 10 presents the statistics across three datasets. We observe

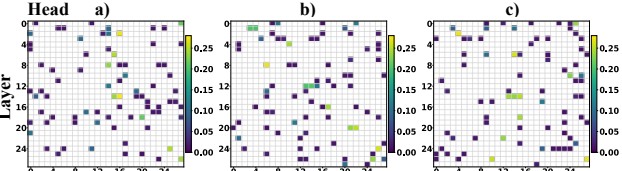

*Figure 3.* Heatmaps of task-sensitive attention heads (10%) selected by P2D on Qwen-2.5-7B-Instruct. The distinct activation patterns across a) GSM8K, b) DialogSum, and c) BioInstruct confirm that different tasks rely on disparate sparse sub-networks.

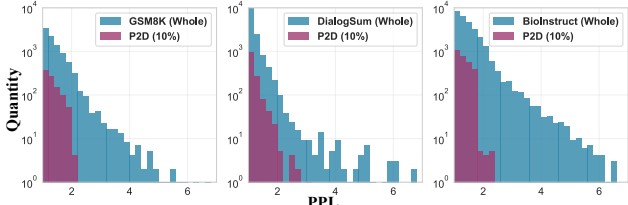

*Figure 4.* Data distribution of all samples vs. the 10% subset extracted by P2D on Qwen-2.5-7B-Instruct. The distinct skew toward lower perplexity signifies High-Affinity: the selected data structurally resonates with the identified task-sensitive heads.

that the average sample lengths (both in tokens and words) of the P2D subset are remarkably close to, and in some cases slightly higher than, those of the full dataset (e.g., 227.0 vs. 231.5 tokens for DialogSum). This length consistency serves as strong evidence that P2D does not exploit length bias. Instead, the lower perplexity indicates High-Affinity: the selected samples possess structural patterns that resonate more strongly with the identified task-sensitive heads ($\mathcal{H}_T$). This confirms the Strong Map Hypothesis: the identified sparse parameters act as a *key* that naturally fits specific data samples (the *lock*). By filtering for this high-affinity data, P2D establishes a synergistic pipeline, thereby maximizing alignment efficiency. More details in Appendix C.

## 6. Conclusion

In this paper, we presented **P2D**, a unified framework grounded in the observation of **Strong Map Hypothesis**: a sparse subset of attention heads plays a dominant role in task-specific adaptation. Unlike approaches that isolate data selection from adaptation, P2D leverages task-sensitive attention heads as a dual compass to establish a robust and **synergistic pipeline**, where structural probes guide sample mining and high-affinity data refines the structure. To rigorously quantify end-to-end costs, we introduced the Alignment Efficiency Ratio (AER). Empirically, P2D updates merely 10% of attention heads on 10% of the data, achieving an 8.3 pp performance gain and a 7.0× speedup over strong baselines. These results validate that unlocking the resonance between model structure and data is a pivotal direction for efficient LLM alignment.

## Acknowledgements

This work is supported by the Fundamental and Interdisciplinary Disciplines Breakthrough Plan of the Ministry of Education of China. This paper is also supported by the National Regional Innovation and Development Joint Fund (No. U24A20254) and the Ant Group Research Fund.

## Impact Statement

This paper presents work whose goal is to advance the field of Efficient Machine Learning. While our method contributes to Green AI and democratizes access by minimizing computational barriers, we recognize that data selection algorithms can inadvertently reflect or amplify biases present in the training data. Additionally, the reduced cost of adaptation could potentially facilitate misuse. We therefore encourage practitioners to conduct rigorous safety and fairness evaluations when deploying such strategies.

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

# A. Experiment Details

## A.1. Datasets

We evaluate our approach on a suite of publicly available benchmarks chosen to cover a variety of task formats and difficulty levels. Table 11 summarizes each task, including the exact sample count and sequence length statistics. To promote fair comparison and reproducibility, we shuffle each dataset with a fixed random seed and split it into 90% training and 10% testing subsets, following the protocol of Wang et al. (2025). This 9:1 split strikes a balance between providing ample data for model adaptation and retaining a sufficiently large held-out set for reliable evaluation of generalization.

To accurately quantify the data complexity, we computed the average token and word counts for each dataset. The tokenization was performed using the `tiktoken` library with the `cl100k_base` encoding scheme (aligned with GPT-4). As detailed in the table, **DialogSum** presents the longest average sequence length (227.03 tokens), reflecting the verbose nature of dialogue tasks, whereas **BioInstruct** (93.50 tokens) and **GSM8K** (157.50 tokens) are relatively more concise.

*Table 11.* Overview of the datasets employed in our experiments. We report the exact sample counts, along with the average number of tokens and words per sample. Token counts are calculated using the `tiktoken` library with the `cl100k_base` encoding.

| Name | Task | # Samples | Avg. Tokens | Avg. Words |
|---|---|---|---|---|
| GSM8K (Cobbe et al., 2021) | Math | 6,725 | 157.50 | 96.75 |
| DialogSum (Chen et al., 2021) | Dialogue | 12,460 | 227.03 | 153.85 |
| BioInstruct (Tran et al., 2024) | Biomedical | 22,504 | 93.50 | 70.92 |

## A.2. Implementation Details

**Software and Hardware.** All models are implemented using PyTorch (Paszke et al., 2019), the HuggingFace Transformers library (Wolf et al., 2019), and the **LLaMA-Factory** framework[1]. For efficient training on large-scale models, we leverage DeepSpeed (Rasley et al., 2020) with ZeRO-2 (Rajbhandari et al., 2020) optimization. Training is conducted on a cluster of 8 NVIDIA A100 (80GB) GPUs.

**Statistical Reproducibility.** To ensure the statistical reliability of our results and mitigate the variance inherent in few-shot data selection and optimization, we conduct all main experiments using three distinct random seeds (42, 43, 44). Unless otherwise stated, the results reported in the main tables (e.g., Table 1, Table 2) are the average performance across these three runs.

**Baseline Implementation Details.** To ensure reproducibility and a rigorous comparison, we align the implementation of all baselines with our unified budget constraints, utilizing their official public codebases where available.

For **LOFIT** (Yin et al., 2024), we utilize the official implementation to calculate module importance based on activation magnitude changes relative to a pre-computed prior. We select the top-$\rho_P$ representations (aligned with our 10% parameter budget) and fine-tune them while freezing the rest. Crucially, this selection uses the same training subset as P2D to ensure fairness. For **LoRA** (Hu et al., 2022), we restrict adaptation to low-rank updates of rank $r = 128$ inserted into the $W_q, W_k, W_v$ projection matrices. This configuration results in a trainable parameter count comparable to our sparse head setting, ensuring that performance differences stem from structural efficiency rather than parameter scale.

For **IFD** (Li et al., 2024a)[2], **Nuggets** (Li et al., 2024b)[3], and **Data Whisperer** (Wang et al., 2025)[4], we employ their respective official GitHub repositories to score and select samples. Specifically, IFD calculates the "Complex-Simple" loss ratio; Nuggets measures the divergence from a one-shot reference distribution; and Data Whisperer performs stratified sampling based on semantic embedding clustering. In all few-shot in-context learning setups for data selection, we follow the protocol of Wang et al. (2025) by concatenating five demonstration examples and three query examples per input. Finally, we assess summarization quality on DialogSum and BioInstruct using ROUGE-L (Lin, 2004), and report Exact Match accuracy on GSM8K.

---

[1]https://github.com/hiyouga/LlamaFactory
[2]https://github.com/tianyi-lab/Cherry_LLM
[3]https://github.com/pldlgb/nuggets
[4]https://github.com/gszfwsb/Data-Whisperer

**Optimization and Hyperparameters.** All models are trained using the AdamW optimizer (Loshchilov, 2017), which decouples weight decay from the gradient update. We set the momentum parameters to $\beta_1 = 0.9$ and $\beta_2 = 0.999$, and apply a weight decay of $0.1$ to regularize the learned weights. To stabilize early training, the learning rate is linearly warmed up over the first 10% of total steps; afterwards, it follows a cosine annealing schedule that decays it smoothly to 10% of its initial value. For the main adaptation stage (including standard SFT and our P2D sparse fine-tuning), we use an effective batch size of 128 examples. Distinctly, for the Fast Head Identification (FHI) phase, we utilize a sub-dataset of only 100 examples. To ensure stable gradient estimation on this small sample, we perform full-batch training (batch size = 100) for 20 steps, which serves solely as a probe to identify structural sensitivity. We initialize the learning rate at $2 \times 10^{-5}$. Each model is trained for three epochs with mixed-precision (FP16) arithmetic on 8 NVIDIA A100 GPUs, which both speeds up training and reduces memory footprint.

---

**Algorithm 1** From Parameters to Data (P2D)

---

**Input**: pretrained model $\mathcal{M}_B$, unlabeled pool $\mathcal{D}$
**Parameter**: head fraction $\rho_P$, data fraction $\rho_D$, demo size $n_d$, query size $n_q$, FHI shots $n_{\text{FHI}}$
**Output**: adapted model $\mathcal{M}_{P2D}$, head set $\mathcal{H}_T$, data subset $\mathcal{D}_T$

  1: **// 1. Fast Head Identification**
  2: Fine-tune $\mathcal{M}_B$ on $n_{\text{FHI}}$ examples $\rightarrow \mathcal{M}_T$
  3: **for** each head $h$ **do**
  4:     Compute composite $W_o^h$; obtain $P_B^h, P_T^h = \text{Softmax}(\text{Flatten}(\boldsymbol{W}_{\text{comp}}^h)/\tau)$
  5:     $s_h \leftarrow W_1(P_B^h, P_T^h)$
  6: **end for**
  7: $\mathcal{H}_T \leftarrow$ top $\rho_P$ heads by $s_h$
  8: **// 2. Parameter-Guided Data Selection**
  9: Initialize $s_i \leftarrow 0, \; n_i \leftarrow 0 \; \forall i \in \mathcal{D}$
10: **for** $t = 1$ to iteration count **do**
11:     Sample demo set $\mathcal{D}_d$ and query set $\mathcal{D}_q$
12:     Compute base score $s = \frac{1}{|\mathcal{D}_q|} \sum f(\hat{y}_q, y_q)$
13:     **for** each demo example $i \in \mathcal{D}_d$ **do**
14:         Compute weight $w_i$ from heads $\mathcal{H}_T$
15:         Update: $s_i + = s \cdot w_i; n_i + = 1$
16:     **end for**
17: **end for**
18: Finalize scores: $s_i \leftarrow s_i / n_i$
19: Select $\mathcal{D}_T \leftarrow$ top $\rho_D$ examples by $s_i$
20: **// 3. Sparse Head Adaptation**
21: For each head $h$, apply $\nabla_{\theta^h} \mathcal{L} \leftarrow m_h \cdot \nabla_{\theta^h} \mathcal{L}$ where $m_h = \mathbb{I}[h \in \mathcal{H}_T]$
22: Fine-tune $\mathcal{M}_B$ on $\mathcal{D}_T$ updating only $\mathcal{H}_T$
23: **return** $\mathcal{M}_{P2D}, \mathcal{H}_T, \mathcal{D}_T$

---

# B. Algorithm

The detailed algorithm of **P2D** is given in Algorithm 1. Starting from a pre-trained base model $\mathcal{M}_0$ and a labeled task dataset $\mathcal{D}$, P2D proceeds in three stages:

1. **Fast Head Identification:** Each attention head in $\mathcal{M}_0$ is scored according to its relevance to the downstream task, using a lightweight proxy model or gradient-based criterion. The top-ranking heads form the task-sensitive set $\mathcal{H}_{\mathcal{T}}$.

2. **Parameter-Guided Data Selection:** We use $\mathcal{H}_{\mathcal{T}}$ to compute a relevance score for every example in $\mathcal{D}$, for instance by measuring how strongly the identified heads activate on each input. The highest-scoring instances are gathered into the curated subset $\mathcal{D}_{\mathcal{T}}$.

3. **Sparse Head Adaptation:** Only the parameters corresponding to $\mathcal{H}_{\mathcal{T}}$ are updated, and training is performed exclusively on $\mathcal{D}_{\mathcal{T}}$. This focused adaptation produces the final model $\mathcal{M}_{P2D}$ while preserving the remaining pretrained weights.

By restricting both the parameter updates and the training data, P2D achieves significant efficiency gains without sacrificing performance. The algorithm outputs the adapted model $\mathcal{M}_{P2D}$, the selected head set $\mathcal{H}_{\mathcal{T}}$, and the curated data subset $\mathcal{D}_{\mathcal{T}}$.

## C. Qualitative Analysis: What does P2D Select?

To better understand the mechanics behind the **Strong Map Hypothesis**, we qualitatively inspect the two core outputs of our pipeline: the task-sensitive attention heads identified by the FHI module, and the high-affinity data samples curated by the parameter-guided selection.

### C.1. Structural Selection: Visual Analysis of Cross-Task Overlap

A critical question regarding the **Strong Map Hypothesis** is whether the identified task-sensitive heads are universal (shared across tasks) or highly specific. To address this, we analyze the structural heatmaps presented in Figure 3, 10 and 11.

Comparing the activation patterns across GSM8K (a), DialogSum (b), and BioInstruct (c), we observe two distinct phenomena that corroborate the task-specificity of our method:

- **Structural Specialization (Visual Divergence):** The "hotspots" (highly sensitive heads) for reasoning tasks like GSM8K are spatially distinct from those for summarization or biomedical tasks. For instance, in Qwen-3-8B (Figure 10), GSM8K relies heavily on specific heads in deeper layers to handle multi-step logic, whereas DialogSum activates a more dispersed set of heads in intermediate layers to manage context aggregation. This visual divergence indicates that the "keys" required to unlock performance are largely task-specific, with minimal functional overlap between disparate domains.

- **Shared Functional Core:** Despite the divergence, we observe a small subset of faint heads that remain active across multiple tasks. We hypothesize these represent "general-purpose" heads responsible for fundamental linguistic competencies (e.g., syntax, coreference resolution) required by all instructions.

In summary, while a foundational linguistic core exists, the **dominant** drivers of downstream performance are spatially distinct across tasks. P2D succeeds by dynamically pinpointing these unique sub-networks rather than relying on a static parameter set.

*Table 12.* Case study of samples from GSM8K. **Selected** samples typically involve complex, multi-stage logic chains (e.g., tracking cumulative progress with interruptions) that strongly activate our identified reasoning heads. In contrast, **Discarded** samples often feature simple linear arithmetic with low structural density.

| Status | Example Content (Real GSM8K Samples) |
|---|---|
| **Selected** | *Question:* Janice can type 6 sentences per minute. She typed for 20 minutes, took a break, and typed 15 minutes longer. She then had to erase 40 sentences she had typed incorrectly. After a meeting, she typed for 18 minutes more. In all, the paper had 536 sentences by the end of today. How many sentences did she start with today? 
 *Answer:* Janice typed $6 \times 20 = 120$ sentences before her break, $6 \times 15 = 90$ sentences after, and $6 \times 18 = 108$ sentences after her meeting. Total typed today: $120 + 90 + 108 = 318$. After erasing 40, she had $318 - 40 = 278$ left. Since the total is 536, she started with $536 - 278 = 258$. #### 258 
 *Analysis:* High complexity. Requires tracking state changes over time, mixing multiplication, subtraction, and reverse-engineering the initial state. Strongly resonates with reasoning heads. |
| **Discarded** | *Question:* A cat has nine lives. A dog has 3 less lives than a cat. A mouse has 7 more lives than a dog. How many lives does a mouse have? 
 *Answer:* Dog: $9 - 3 = 6$ lives. Mouse: $6 + 7 = 13$ lives. #### 13 
 *Analysis:* Low complexity. Simple sequential addition and subtraction. Lacks the structural depth to trigger the sparse "winning ticket" sub-network. |

### C.2. Data Selection: High-Affinity vs. Low-Affinity Samples

We observe that samples selected by P2D (High-Affinity) often exhibit clear, structured reasoning steps that align with the specific "Arithmetic Reasoning" heads identified in the model. For instance, questions involving multi-step algebra tend to activate the identified heads strongly. Conversely, discarded samples (Low-Affinity) often contain linguistic ambiguities or require external knowledge not centered on pure reasoning, which fails to resonate with the targeted sparse structure. This aligns with the Lottery Ticket Hypothesis (Frankle & Carbin, 2018), suggesting that P2D identifies the "winning ticket" data that matches the "winning ticket" sub-network.

## D. Analysis of the FHI

*Table 13.* Ablation study on FHI hyperparameters. We report the final P2D performance (using **10% heads**) based on heads identified under different settings. The default setting (100 Ex. / 20 Steps) achieves the optimal trade-off.

| FHI Settings | | Final P2D Performance | | |
| --- | --- | --- | --- | --- |
| Examples | Steps | GSM8K | DialogSum | BioInstruct |
| 50 | 10 | 75.20 | 41.50 | 29.80 |
| 50 | 20 | 76.45 | 42.10 | 31.20 |
| 100 | 10 | 77.10 | 42.80 | 32.10 |
| **100** | **20** | **78.82** | **44.04** | **32.74** |
| 100 | 50 | 78.85 | 44.08 | 32.79 |

In our Fast Head Identification (FHI) phase, we utilize a lightweight proxy model trained with limited data (100 examples) and steps (20 steps) to compute the matrix distance for head importance ranking. A natural question arises: *Are these hyperparameters sufficient to locate the truly important heads?* To investigate this, we conducted an ablation study by varying the number of training examples {50, 100} and update steps {10, 20, 50}. We evaluated the quality of the identified heads by plugging them into the final P2D training pipeline and measuring the downstream performance on GSM8K and DialogSum. As illustrated in Table 13, increasing the number of examples from 50 to 100 yields a noticeable performance gain, indicating that extremely sparse data (50 samples) might introduce noise into the importance estimation. However, further increasing the training steps from 20 to 50 provides negligible improvements (e.g., 78.82 vs. 78.85 on GSM8K) while linearly increasing the FHI cost. We observe that the importance distribution of attention heads stabilizes very early in the training process. Therefore, we adopt the setting of 100 examples and 20 steps as a cost-effective sweet spot for all our experiments.

**Sensitivity to Softmax Temperature ($\tau$).** In Equation (6), the temperature $\tau$ controls the sharpness of the parameter distribution. We empirically set $\tau = 0.1$. Larger $\tau$ values ($\tau > 1$) lead to overly uniform distributions, diluting the signal of task-specific heads (approximating random selection). Conversely, extremely small $\tau$ values approximate an *argmax* operation, potentially missing secondary but supportive heads. This behavior mimics the calibration effects observed in knowledge distillation (Hinton, 2015).

**Proxy Steps: 20 vs. 50.** Table 13 already shows that extending the proxy from 20 to 50 steps yields negligible improvement on GSM8K. To stress-test this, we also examined a 50-step variant on DialogSum/BioInstruct: GSM8K $78.82 \rightarrow 79.37$, DialogSum $44.04 \rightarrow 42.18$, BioInstruct $32.74 \rightarrow 33.61$. Notably, DialogSum slightly *regresses* with the 50-step proxy, even though the proxy's own task accuracy jumps from 18.45 to 52.78 on GSM8K. This empirically dissociates *structural sensitivity* (which stabilizes within 20 steps) from *task accuracy* (which keeps improving). The proxy is a structural probe, not a solver, and 20 steps suffice. Adding more steps merely inflates the FHI cost without sharpening the head ranking.

## E. Sensitivity Scoring: $W_1$ vs. Alternative Metrics

A natural alternative to the Wasserstein-1 ($W_1$) distance used in Eq. 6 is to score heads via activation-gradient information (e.g., Fisher / SNIP-style metrics). We provide both a theoretical argument and an empirical comparison across scales.

**Theoretical Argument: Linear vs. Quadratic Suppression.** After only $T = 20$ proxy steps, the parameter drift $\Delta = \theta_T^h - \theta_B^h$ is infinitesimal in the spectral sense ($\|\Delta\| \ll 1$). Treating each head's softmax-flattened weight as a distribution $P^h$, a small parameter translation $P_T^h \approx P_B^h(x - \Delta)$ implies:

$$W_1(P_B^h, P_T^h) \approx \|\Delta\| \qquad \text{(linear)}$$
$$D_{GB}(\pi_B^h \| \pi_T^h) \approx \tfrac{1}{2}\mathcal{I}(\pi_B^h)\|\Delta\|^2 \qquad \text{(quadratic)}$$

where $\mathcal{I}$ is the Fisher information. The quadratic suppression in gradient-based scoring disproportionately buries subtle but critical structural drifts into noise; $W_1$'s linear sensitivity preserves them. $W_1$ is also *data-free* (no forward/backward passes), while gradient metrics require a calibration batch.

**Computational Efficiency: $W_1$ Sits in the Sweet Spot.** Beyond the theoretical argument above, $W_1$ also dominates alternative scoring metrics on the practical quality–cost trade-off. Table 14 compares $W_1$ against (i) a cheap baseline (cosine similarity over the same softmax-flattened weight matrices) and (ii) the expensive gradient-based metric of the theoretical comparison, all on Qwen-2.5-7B with the standard $10\%/10\%$ P2D pipeline. The cheap baseline collapses GSM8K to 63.28 (vs. $W_1$'s 78.82): its uniform geometry is insufficient to distinguish task-sensitive heads from background drift. The gradient-based scorer recovers most of $W_1$'s quality but costs $\sim 10\,\text{min}$, more than $20\times$ the $\sim 30\,\text{s}$ incurred by $W_1$. Critically, $W_1$ matches or exceeds the dynamic metric on two of the three benchmarks while spending only $\sim 3\%$ of the scoring budget, and unlike the gradient metric, it requires no calibration batch.

*Table 14.* Head-scoring metrics on Qwen-2.5-7B (standard $10\%/10\%$ P2D pipeline). $W_1$ matches the expensive gradient metric in quality and the cheap cosine metric in cost, providing the best practical operating point.

| Metric | GSM8K | DialogSum | BioInstruct | Scoring Time |
|---|---|---|---|---|
| Cosine similarity | 63.28 | 28.37 | 21.43 | $\sim 10$ s |
| Gradient-based | 76.48 | 43.79 | **33.08** | $\sim 10$ min |
| $W_1$ **(Ours)** | **78.82** | **44.04** | 32.74 | $\sim 30$ s |

**Empirical Comparison Across Scales.** Table 15 reports head identified by $W_1$ vs. gradient-based scoring on Qwen-2.5 at 7B, 14B, and 32B. $W_1$'s advantage *widens* with scale: BioInstruct, where the 7B comparison was slightly in favor of the gradient metric ($-0.34$ pp), reverses to $+0.92$ at 14B and $+2.00$ at 32B. Importantly, the scoring-time gap also grows by orders of magnitude (gradient: $\sim 10\,\text{min}\rightarrow\sim 30\,\text{min}$ on domain data; $\sim 1.9\,\text{h}\rightarrow\sim 5.7\,\text{h}$ on generalist data) while $W_1$ remains within a few minutes regardless of model or dataset size.

*Table 15.* $W_1$ vs. gradient-based head scoring across 7B/14B/32B Qwen-2.5-Instruct backbones. Scoring time is reported as (domain task / generalist UltraChat). Performance is downstream after the standard $10\%/10\%$ P2D pipeline. $W_1$'s advantage compounds with model scale on all benchmarks.

| Model | Method | Scoring Time (domain / generalist) | GSM8K | DialogSum | BioInstruct | MMLU | ARC-C |
|---|---|---|---|---|---|---|---|
| 7B | Gradient-based | $\sim 10$ min / $\sim 1.9$ h | 76.48 | 43.79 | **33.08** | 66.85 | 76.92 |
| | $W_1$ (Ours) | $\sim 30$ s / $\sim 30$ s | **78.82** | **44.04** | 32.74 | **67.45** | **78.23** |
| 14B | Gradient-based | $\sim 23$ min / $\sim 4.3$ h | 86.55 | 48.21 | 36.92 | 72.14 | 82.35 |
| | $W_1$ (Ours) | $\sim 70$ s / $\sim 70$ s | **87.62** | **49.38** | **37.84** | **73.88** | **84.12** |
| 32B | Gradient-based | $\sim 30$ min / $\sim 5.7$ h | 88.65 | 51.34 | 44.23 | 75.95 | 85.13 |
| | $W_1$ (Ours) | $\sim 100$ s / $\sim 100$ s | **89.34** | **53.45** | **46.23** | **78.45** | **87.23** |

# F. Scaling Analysis Across Model Sizes

We extend the main-text scaling evaluation (Section 5, Table 5) by reporting the full per-task results across Qwen-2.5 at four scales: 1.5B, 3B, 14B, and 32B. In each table, we vary both the data-selection strategy (rows: Full Dataset, Data Whisperer, P2D$^\dagger$) and the parameter-update strategy (columns: Full Params, LoRA, P2D$^\ddagger$). Throughout, P2D$^\dagger$ uses our parameter-guided data selection with $\rho_D = 10\%$, and P2D$^\ddagger$ uses sparse head adaptation with $\rho_P = 10\%$.

*Table 16.* Full per-task results on **Qwen-2.5-1.5B-Instruct**. Rows vary the data-selection strategy ($\rho_D = 10\%$ for P2D$^\dagger$ and Data Whisperer; 100% for Full Dataset); columns vary the parameter-update strategy. At 1.5B, GSM8K shows positive gains, while domain tasks lag slightly as smaller models require more data to saturate domain knowledge.

| Data ($\rho_D$) | Params ($\rho_P$) | GSM8K | DialogSum | BioInstruct |
|---|---|---|---|---|
| Full Dataset | Full Params | 63.45 | 29.34 | 18.56 |
| | LoRA | 62.18 | 30.07 | 17.93 |
| | P2D$^\ddagger$ | 65.23 | 31.48 | 19.74 |
| Data Whisperer | Full Params | 61.34 | 26.51 | 15.62 |
| | LoRA | 60.72 | 26.83 | 15.18 |
| | P2D$^\ddagger$ | 62.47 | 27.69 | 16.41 |
| P2D$^\dagger$ | Full Params | 62.53 | 27.31 | 16.85 |
| | LoRA | 61.86 | 26.94 | 16.27 |
| | **P2D$^\ddagger$** | **63.81** | **28.46** | **17.63** |

*Table 17.* Full per-task results on **Qwen-2.5-3B-Instruct**. Same column / row conventions as Table 16. At 3B, P2D$^\ddagger$ already surpasses Full SFT on GSM8K; the 10%/10% pipeline narrows the gap on domain tasks compared to 1.5B.

| Data ($\rho_D$) | Params ($\rho_P$) | GSM8K | DialogSum | BioInstruct |
|---|---|---|---|---|
| Full Dataset | Full Params | 72.56 | 36.78 | 24.63 |
| | LoRA | 71.83 | 37.51 | 23.87 |
| | P2D$^\ddagger$ | 74.61 | 38.19 | 25.74 |
| Data Whisperer | Full Params | 70.27 | 33.14 | 21.19 |
| | LoRA | 69.64 | 33.62 | 20.58 |
| | P2D$^\ddagger$ | 71.48 | 34.37 | 22.31 |
| P2D$^\dagger$ | Full Params | 71.19 | 34.83 | 21.76 |
| | LoRA | 70.53 | 34.28 | 21.33 |
| | **P2D$^\ddagger$** | **72.94** | **35.92** | **23.93** |

*Table 18.* Full per-task results on **Qwen-2.5-14B-Instruct**. Same column / row conventions as Table 16. At 14B, the 10%/10% P2D pipeline (last row) *surpasses* Full SFT on all three tasks, confirming the Strong Map becomes more structurally concentrated at larger scale.

| Data ($\rho_D$) | Params ($\rho_P$) | GSM8K | DialogSum | BioInstruct |
|---|---|---|---|---|
| Full Dataset | Full Params | 86.47 | 50.93 | 43.12 |
| | LoRA | 85.36 | 52.14 | 42.56 |
| | P2D$^\ddagger$ | 88.21 | 53.47 | 45.89 |
| Data Whisperer | Full Params | 84.13 | 46.78 | 33.87 |
| | LoRA | 82.56 | 46.95 | 31.94 |
| | P2D$^\ddagger$ | 85.34 | 47.23 | 35.21 |
| P2D$^\dagger$ | Full Params | 85.18 | 48.45 | 35.93 |
| | LoRA | 84.67 | 48.12 | 35.56 |
| | **P2D$^\ddagger$** | **87.62** | **49.38** | **37.84** |

**Summary of Scaling Trends.** Two consistent findings emerge. **First**, P2D$^\ddagger$ (10% heads, full data) outperforms Full SFT at every scale across all three benchmarks, confirming sparse head adaptation is strictly more efficient regardless of model size. **Second**, the data-efficiency advantage of the joint P2D$^\dagger$+P2D$^\ddagger$ pipeline grows monotonically with scale. At 1.5B/3B, GSM8K shows positive gains while DialogSum/BioInstruct lag slightly (smaller models require more data to saturate domain knowledge). At 14B/32B, the 10%/10% pipeline *surpasses* Full SFT on all three tasks, demonstrating that the Strong Map becomes more concentrated and data-efficient at a larger scale. Combined with $W_1$'s near-constant scoring

*Table 19.* Full per-task results on **Qwen-2.5-32B-Instruct**. Same column / row conventions as Table 16. The 10%/10% P2D pipeline yields the largest gains over Full SFT at this scale, and $W_1$'s scoring overhead remains constant at $\sim$100 s (Table 15), making P2D's practical advantage most pronounced at industrial-deployment scales.

| Data ($\rho_D$) | Params ($\rho_P$) | GSM8K | DialogSum | BioInstruct |
|---|---|---|---|---|
| Full Dataset | Full Params | 88.94 | 53.12 | 45.78 |
| | LoRA | 87.73 | 54.38 | 45.29 |
| | P2D$^{\ddagger}$ | 90.23 | 55.34 | 47.89 |
| Data Whisperer | Full Params | 87.45 | 49.17 | 38.81 |
| | LoRA | 86.62 | 49.53 | 37.46 |
| | P2D$^{\ddagger}$ | 88.34 | 50.28 | 40.17 |
| P2D$^{\dagger}$ | Full Params | 88.07 | 52.43 | 41.35 |
| | LoRA | 87.56 | 51.87 | 41.62 |
| | **P2D$^{\ddagger}$** | **89.34** | **53.45** | **46.23** |

overhead (Table 15), P2D's practical advantage compounds substantially at the model sizes most relevant to industrial deployment.

## G. Baseline Fairness: LoRA Rank Search

To address baseline-fairness concerns, we conduct a rank search for LoRA under the same $\rho_D = 10\%$ constraint, on Qwen-2.5-7B-Instruct. We sweep rank $r \in \{32, 64, 128, 256\}$, with the corresponding trainable-parameter ratio shown in the second column. P2D's 10% sparse-head configuration uses approximately $0.94\%$ trainable parameters on this backbone.

*Table 20.* LoRA rank search under 10% data on Qwen-2.5-7B. Even the best-tuned LoRA configuration underperforms P2D's sparse head adaptation while requiring comparable or larger trainable parameter budgets.

| Method | Trainable Params (%) | GSM8K | DialogSum | BioInstruct |
|---|---|---|---|---|
| LoRA ($r = 32$) | 0.24% | 71.83 | 39.42 | 28.51 |
| LoRA ($r = 64$) | 0.48% | 73.52 | 40.27 | 29.86 |
| LoRA ($r = 128$) | 0.95% | 75.18 | 41.34 | 30.42 |
| LoRA ($r = 256$) | 1.90% | 75.42 | 41.65 | 30.81 |
| **P2D (10% heads)** | **0.94%** | **78.82** | **44.04** | **32.74** |

Across all ranks, LoRA underperforms P2D by 3–4 pp on every benchmark, despite occupying a comparable or larger trainable-parameter budget. Note that increasing $r$ from 128 to 256 yields diminishing returns ($+0.24$ pp on GSM8K) at $2\times$ the cost, indicating that the gap is structural rather than capacity-bounded: LoRA's low-rank decomposition cannot exploit the task-specific head topology that P2D directly identifies.

## H. Synergy Validation: Lock-and-Key Cross-Ablation

To verify that P2D is more than a mere orchestration of existing data-selection and PEFT methods, we conduct a strict cross-ablation on Qwen-2.5-7B-Instruct. If P2D were simply a pipeline of replaceable components, swapping the data or the headset with a strong alternative should yield comparable results. Table 21 refutes this hypothesis: pairing the P2D-selected data with *randomly* chosen heads collapses GSM8K performance from 78.82 to 67.13; symmetrically, training the P2D-identified heads on random or IFD/Nuggets/DW-curated data also under-performs the full pipeline by a large margin. Only the *precise pairing* of P2D data with P2D heads achieves peak performance across all three tasks. This empirically validates the Strong Map Hypothesis as a **Lock-and-Key Synergy**: the identified sparse heads (the "lock") and the curated high-affinity data (the "key") are mutually informed and *jointly* necessary, with neither direction alone sufficient.

*Table 21.* Lock-and-Key cross-ablation on Qwen-2.5-7B-Instruct. Each row reports a specific pairing of (data subset, head subset). The synergy between P2D-selected data and P2D-identified heads is irreplaceable.

| Data | Heads | GSM8K | DialogSum | BioInstruct |
|---|---|---|---|---|
| P2D | Random | 67.13 | 38.77 | 25.74 |
| Random | P2D | 72.32 | 40.33 | 27.98 |
| IFD | P2D | 71.42 | 36.06 | 23.82 |
| Nuggets | P2D | 68.91 | 35.07 | 25.91 |
| Data Whisperer | P2D | 75.84 | 42.12 | 30.19 |
| **P2D** | **P2D** | **78.82** | **44.04** | **32.74** |

## I. Disentangling the End-to-End Speedup and ZeRO-2 Communication

**Disentangling Parameter vs. Data Sparsity.** A reasonable concern is whether P2D's $7.0\times$ end-to-end speedup is merely an artifact of training on 10% of the data. Table 22 ablates parameter sparsity and data sparsity independently: updating only 10% of attention heads on the full dataset already yields $\sim 1.72\times$ speedup over Full SFT; selecting 10% of data with full parameters yields $\sim 2.22\times$. Together, with P2D's near-zero selection overhead (Table 2), they compound into the $7.0\times$ end-to-end gain.

*Table 22.* Disentangling the end-to-end speedup: parameter sparsity and data sparsity each contribute substantial, comparable gains. AER lower is better. Speedup is reported relative to Full SFT, averaged across the three datasets on Qwen-2.5-7B-Instruct.

| Setting | Data | Heads | AER↓ / Speedup↑ |
|---|---|---|---|
| Full SFT | 100% | 100% | 1.00 / 1.00× |
| Param Sparsity only | 100% | 10% | 0.58 / ∼1.72× |
| Data Sparsity only | 10% | 100% | 0.45 / ∼2.22× |
| **P2D (both)** | **10%** | **10%** | **0.32 / 7.0× (vs. Nuggets)** |

**ZeRO-2 Communication Breakdown.** We complement the speedup breakdown above with a strict per-step traffic accounting under DeepSpeed ZeRO-2 (Rajbhandari et al., 2020). Each step involves two communication collectives: (i) **Reduce-Scatter** to synchronize gradients ($\sim P_{train} \times 2$ bytes per GPU), and (ii) **All-Gather** to broadcast updated parameters ($\sim P_{train} \times 2$ bytes per GPU). Total per-GPU communication per step is $\sim 2\,P_{train} \times 2$ bytes (under bf16).

For Qwen-2.5-7B under Full SFT ($P_{train} \approx 7.6$B):

$$\text{ZeRO-2 traffic} = 2 \times 7.6\text{B} \times 2 \text{ bytes} \approx 30.4\,\text{GB/step/GPU}.$$

P2D freezes 99% of parameters, leaving $\rho_{train} \approx 0.94\%$ for Qwen-2.5-7B. Hence, P2D's traffic becomes:

$$0.94\% \times 30.4\,\text{GB} \approx 0.30\,\text{GB/step/GPU},$$

a **100× reduction** in per-step synchronization payload. In distributed training, where the gradient-synchronization step typically dominates wall-clock time at scale, this reduction translates directly into wall-clock speedup. The advantage grows with cluster size: as GPU count rises, Full SFT's payload scales linearly with the number of all-reduce participants, while P2D's payload remains negligibly small. AER measurements at different cluster configurations confirm this trend: $0.62 \to 0.54$ when scaling from 4×A100 to 8×A100, and $0.54 \to 0.48$ when migrating from A100 to H200. The 100× communication reduction is therefore not theoretical: it directly underwrites the empirical speedup that is otherwise unexplainable by mere data reduction.

## J. Prompts

For data selection prompts used in our experiments, we follow the settings from Wang et al. (2025). Each prompt is composed of three parts:

- A concise task instruction that states the objective (e.g., "Solve the following math problem:" or "Summarize the dialogue below:").

- A fixed set of demonstration examples formatted as input–output pairs, chosen to cover a range of difficulty and typical patterns for the task.

- A placeholder for the query instance, preceded by the same cue used in the demonstrations (e.g., "Question: ... Answer:" or "Dialogue: ... Summary:").

Figures 5, 6, and 7 show the complete templates for GSM8K, DialogSum, and BioInstruct, respectively, including the final response cue that signals the model to generate its answer. By maintaining a uniform prompt structure while tailoring content to each dataset's domain, we ensure that any performance differences arise from the data selection mechanism itself rather than prompt formatting variations.

## K. Detailed Efficiency Analysis

In Table 23, we provide a granular breakdown of the efficiency and performance across different data selection and training strategies. We report the **Total AER** as the sum of Data Selection time ($T_{sel}$) and Gradient Update time ($T_{train}$), normalized by the full fine-tuning baseline. We also denote the data density ($\rho_D$) and parameter density ($\rho_P$) for each setting. The results highlight three key observations:

- **Minimal Selection Overhead:** **P2D**'s selection phase is computationally negligible ($T_{sel} \approx 0.15$) compared to gradient-based selectors like Nuggets ($T_{sel} \approx 1.46$) or IFD ($T_{sel} \approx 0.63$), ensuring that the pre-processing step does not become a bottleneck.

- **Sparse Training Efficiency:** When equipping data selection methods with our P2D sparse training (updating only 10% of attention heads), we observe a significant reduction in training AER compared to LoRA and standard fine-tuning, reflecting the reduced computational graph depth and memory footprint.

- **Holistic Superiority:** The full **P2D** pipeline (P2D selection + P2D training) achieves the lowest Total AER while consistently outperforming full-parameter baselines, demonstrating that our method effectively identifies and refines the most critical "parameters-to-data" pairs.

## L. Extended Sparsity-Ratio Comparison Tables and Visualizations

In the main text, we reported results only for the 10% data-selection and 10% head-update regimes. Here, we extend those comparisons to include 30%, 50%, and 70% settings, presenting full side-by-side results in Table 24–26. Likewise, whereas we originally showed heatmaps and data-distribution plots for a single model, we now provide the same visualizations for all models. Figures 10–11 display head-importance heatmaps, and Figures 8–9 illustrate the corresponding data-selection distributions for each model. This expanded overview highlights how efficiency and selection patterns evolve as we vary the fraction of data or parameters adapted across different architectures.

## M. Limitations

We outline five limitations of P2D that motivate future work. First, since adaptation is restricted to attention heads while all MLP layers remain frozen, P2D is most effective when downstream tasks primarily require routing or eliciting pre-existing knowledge; tasks that demand injecting genuinely new factual knowledge absent from pretraining may suffer from this MLP bottleneck, and a hybrid that selectively unlocks a small number of MLP components is a natural extension we leave for future work. Second, the Fast Head Identification proxy relies on a 100-sample toy training run, which can in principle miss task-sensitive heads whose signal only emerges with longer training, especially on highly domain-shifted tasks far from the pretraining distribution. Third, the reported $7.0\times$ end-to-end speedup is mmeasured under a specific ZeRO-2 setup on eight A100 GPUs, and the relative contributions of parameter and data sparsity will depend on the optimizer state size, gradient accumulation policy, and inter-GPU bandwidth; we therefore restrict the speedup claim to our reported setting and leave a systematic study across optimizers (e.g., ZeRO-3, FSDP) and hardware tiers to future work. Fourth, the ICL-based data scoring inherits a known sensitivity to demonstration choice (Brown et al., 2020; Min et al., 2022), which P2D mitigates via cross-query demonstration resampling but does not eliminate; a fully demonstration-free scoring procedure is an attractive direction. Finally, P2D is evaluated on supervised fine-tuning; extending it to reinforcement learning alignment (e.g., RLHF, DPO) and exploring a P2D-LoRA hybrid that composes sparse head adaptation with low-rank updates are promising directions for further work.

**GSM8K**

**Instruction:**
You are an expert math assistant. Your role is to provide step-by-step calculations for each problem and deliver the correct final answer. Each solution should be logically structured, with no extra commentary or deviation from the required steps. Your responses must be concise, accurate, and in the exact format specified below. Your sole focus should be on solving the problem as efficiently as possible. Do not include any extraneous information.
### Guidelines for your response:
1. Your response must contain only step-by-step calculations and the final answer.
2. The final output **must** be formatted as: #### <number>.
Replace '<number>' with the correct final result (either an integer or a floating-point number). No deviations or alternative formats are allowed.
3. Do not add any commentary, questions, greetings, or extra remarks.
4. Ensure your calculations are clear, concise, and correct, but only include the steps required to arrive at the final answer.
Please answer each question step by step and provide the final answer following the instructions below.
**Input:**
Below are some demonstrations of how to format your answers:
Question: <Question 1>
Answer: <Answer 1>
...
**Strictly use the format specified below:**
Question 1 Answer: <your step-by-step solution>
#### <final answer>
Question 2 Answer: <your step-by-step solution>
#### <final answer>
(and so on...).
#### Now, based on the provided questions, respond to the following mathematical problems:
Question 1: <query 1>
Question 2: <query 2>
...

*Figure 5.* Parameter-Guided data selection prompt for GSM8K dataset

**DialogSum**

**Instruction:**
You are an expert assistant. Your task is to provide clear, concise, and complete summaries for the given dialogues. Your summaries should accurately capture the main points of each dialogue. Avoid unnecessary details and ensure clarity.
### Guidelines for your response:
1. **Summarize the dialogue concisely and fully**, ensuring all main points are captured.
2. **Avoid adding extra commentary or irrelevant details** that are not part of the dialogue content..
3. If a dialogue is unclear, incomplete, or lacks meaningful content, respond with "No valid content to summarize.
4. **Ensure every summary field is filled out.** Leaving any field blank is not allowed.
Please summarize dialogues based on the given instructions and demonstrations below.
**Input:**
Below are some demonstrations of how to format your answers:
Dialogue: <Dialogue 1>
Summary: <Summary 1>
...
**Strictly use the format specified below:**
Summary 1: <Your summary to Dialogue 1.>
Summary 2: <Your summary to Dialogue 2.>
(and so on...).
#### Now, based on the provided dialogues, provide concise and complete summaries for the following dialogues:
Dialogue 1: <dialogue content>
Question 2: <dialogue content>
...

*Figure 6.* Parameter-Guided data selection prompt for DialogSum dataset

---

**BioInstruct**

**Instruction:**
You are a medical expert. Given an input and an instruction, your objective is to respond with the correct and concise answer based on the provided context.
### Guidelines for your response:
1. **Ensure your responses are concise, clear, and focused on the provided instruction.**
2. **Follow the logical order of questions.** Do not skip or merge responses.
3. **Avoid adding extra commentary or irrelevant details. DO NOT repeat or summarize the question.**
**Input:**
Below are some demonstrations of how to format your answers:
Instruction: <Demonstration 1 Instruction>
Input: <Demonstration 1 Input>
Answer: <Demonstration 1 Answer>
...
**Strictly use the format specified below:**
Question 1 Answer: <your answer to Question 1.>
Question 2 Answer: <your answer to Question 2.>
(and so on...).
#### Now, based on the biomedical demonstrations provided, respond to the following biomedical questions:
Question 1: <query 1>
Question 2: <query 2>
...

---

*Figure 7.* Parameter-Guided data selection prompt for BioInstruct dataset

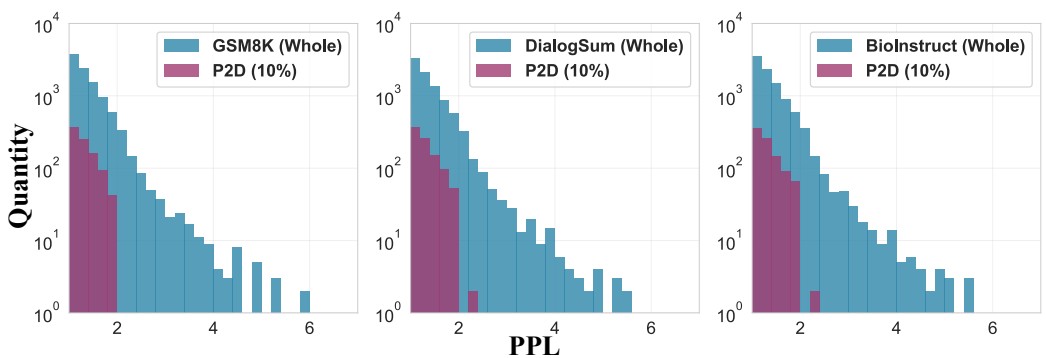

*Figure 8.* Data distribution of all samples and 10% samples extracted by our method. PPL stands for perplexity and is calculated with Qwen-3-8B.

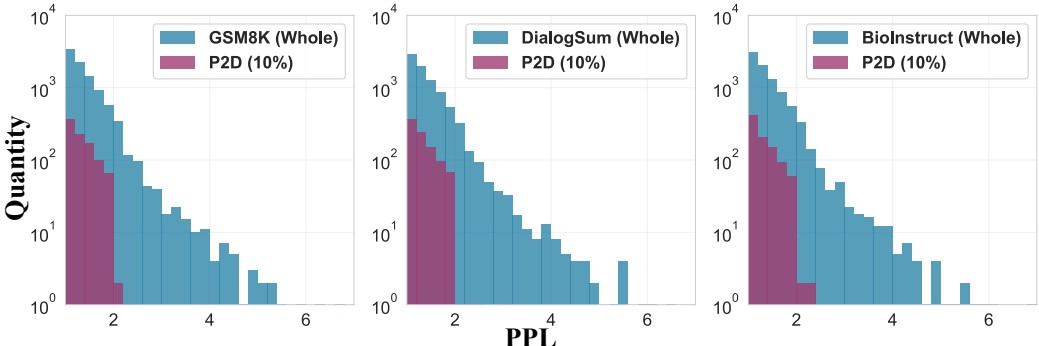

*Figure 9.* Data distribution of all samples and 10% samples extracted by our method. PPL stands for perplexity and is calculated with Llama-3-8B-Instruct.

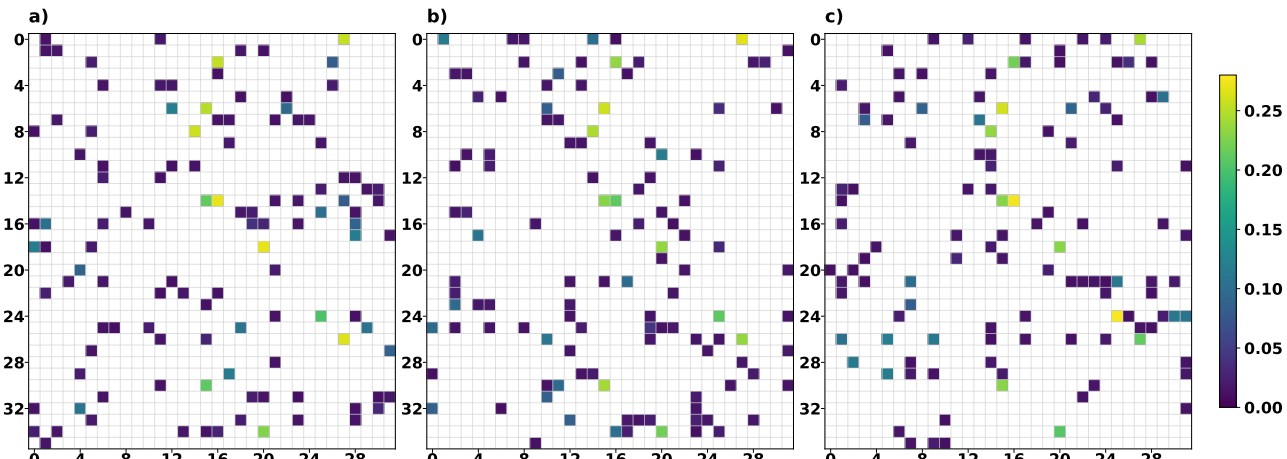

*Figure 10.* Heatmap of attention heads on Qwen3-8B for **a)** GSM8K, **b)** DialogSum, and **c)** BioInstruct datasets. The color intensity indicates the sensitivity of each head.

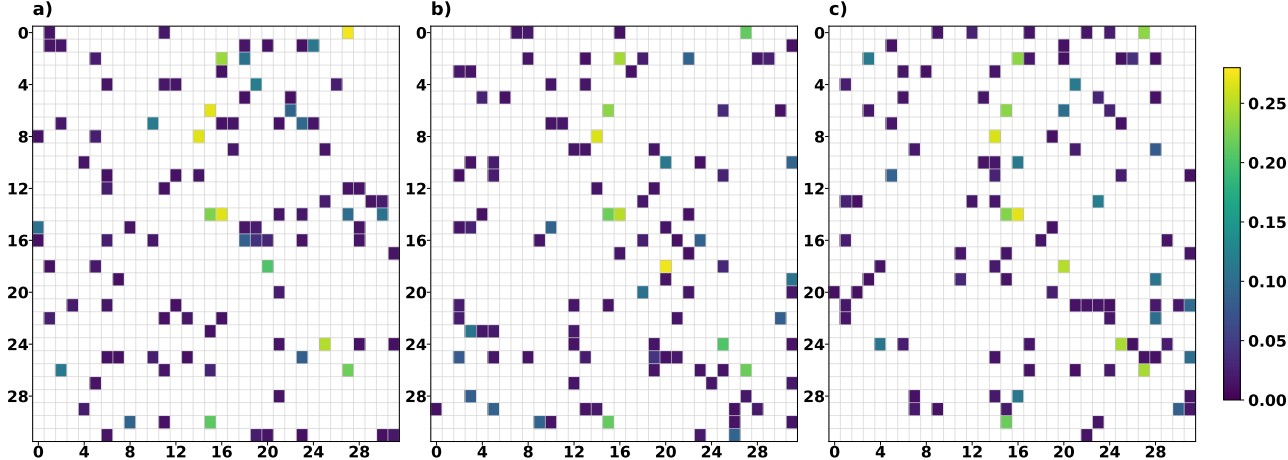

*Figure 11.* Heatmap of attention heads on Llama-3-8B-Instruct for **a)** GSM8K, **b)** DialogSum, and **c)** BioInstruct datasets. The color intensity indicates the sensitivity of each head.

*Table 23.* Comprehensive breakdown of Efficiency Ratio (AER) and Performance. **Bold** values indicate the best results (lowest AER or highest Performance) within each method block.

| Method | $\rho_D$ | $\rho_P$ | GSM8K | | DialogSum | | BioInstruct | |
|---|---|---|---|---|---|---|---|---|
| | | | AER (Sel.+Tr.)↓ | Perf.↑ | AER (Sel.+Tr.)↓ | Perf.↑ | AER (Sel.+Tr.)↓ | Perf.↑ |
| ***Full (full dataset)*** | | | | | | | | |
| + full parameters | 100% | 100% | 1.00 (0.00+1.00) | 78.36 | 1.00 (0.00+1.00) | 46.49 | 1.00 (0.00+1.00) | 39.25 |
| + random | 100% | 10% | **0.49** (0.00+0.49) | 65.63 | **0.49** (0.00+0.49) | 40.52 | **0.49** (0.00+0.49) | 33.75 |
| + LoRA | 100% | 10% | 0.60 (0.00+0.60) | 76.67 | 0.60 (0.00+0.60) | 46.66 | 0.60 (0.00+0.60) | 39.42 |
| + LoFiT | 100% | 10% | 0.56 (0.00+0.56) | 75.60 | 0.56 (0.00+0.56) | 45.78 | 0.56 (0.00+0.56) | 38.56 |
| + *P2D‡ (Ours)* | 100% | 10% | 0.54 (0.00+0.54) | **79.55** | 0.54 (0.00+0.54) | **48.24** | 0.54 (0.00+0.54) | **40.87** |
| ***IFD*** | | | | | | | | |
| + full parameters | 10% | 100% | 1.08 (0.63+0.45) | 69.54 | 1.18 (0.73+0.45) | 35.46 | 1.12 (0.67+0.45) | **27.19** |
| + random | 10% | 10% | **0.75** (0.63+0.12) | 61.89 | **0.85** (0.73+0.12) | 31.10 | **0.79** (0.67+0.12) | 22.60 |
| + LoRA | 10% | 10% | 0.87 (0.63+0.24) | 68.18 | 0.97 (0.73+0.24) | 33.26 | 0.91 (0.67+0.24) | 23.94 |
| + LoFiT | 10% | 10% | 0.82 (0.63+0.19) | 67.27 | 0.92 (0.73+0.19) | 32.38 | 0.86 (0.67+0.19) | 23.12 |
| + *P2D‡ (Ours)* | 10% | 10% | 0.80 (0.63+0.17) | **71.42** | 0.90 (0.73+0.17) | **36.06** | 0.84 (0.67+0.17) | 23.82 |
| ***Nuggets*** | | | | | | | | |
| + full parameters | 10% | 100% | 1.91 (1.46+0.45) | 68.23 | 3.28 (2.83+0.45) | 34.87 | 1.62 (1.17+0.45) | **27.51** |
| + random | 10% | 10% | **1.58** (1.46+0.12) | 61.88 | **2.95** (2.83+0.12) | 31.60 | **1.29** (1.17+0.12) | 23.48 |
| + LoRA | 10% | 10% | 1.70 (1.46+0.24) | 66.67 | 3.07 (2.83+0.24) | 34.72 | 1.41 (1.17+0.24) | 27.30 |
| + LoFiT | 10% | 10% | 1.65 (1.46+0.19) | 65.75 | 3.02 (2.83+0.19) | 33.56 | 1.36 (1.17+0.19) | 26.45 |
| + *P2D‡ (Ours)* | 10% | 10% | 1.63 (1.46+0.17) | **68.91** | 3.00 (2.83+0.17) | **35.07** | 1.34 (1.17+0.17) | 25.91 |
| ***Data Whisperer*** | | | | | | | | |
| + full parameters | 10% | 100% | 0.88 (0.43+0.45) | 74.58 | 1.12 (0.67+0.45) | 41.52 | 0.77 (0.32+0.45) | 29.90 |
| + random | 10% | 10% | **0.55** (0.43+0.12) | 66.09 | **0.79** (0.67+0.12) | 36.12 | **0.44** (0.32+0.12) | 24.19 |
| + LoRA | 10% | 10% | 0.67 (0.43+0.24) | 73.67 | 0.91 (0.67+0.24) | **42.37** | 0.56 (0.32+0.24) | 28.75 |
| + LoFiT | 10% | 10% | 0.62 (0.43+0.19) | 72.45 | 0.86 (0.67+0.19) | 41.08 | 0.51 (0.32+0.19) | 27.71 |
| + *P2D‡ (Ours)* | 10% | 10% | 0.60 (0.43+0.17) | **75.84** | 0.84 (0.67+0.17) | 42.12 | 0.49 (0.32+0.17) | **30.19** |
| ***P2D† (Ours)*** | | | | | | | | |
| + full parameters | 10% | 100% | 0.60 (0.15+0.45) | 76.32 | 0.66 (0.21+0.45) | 43.30 | 0.53 (0.08+0.45) | 32.12 |
| + random | 10% | 10% | **0.27** (0.15+0.12) | 67.13 | **0.33** (0.21+0.12) | 38.77 | **0.20** (0.08+0.12) | 25.74 |
| + LoRA | 10% | 10% | 0.39 (0.15+0.24) | 75.82 | 0.45 (0.21+0.24) | 42.86 | 0.32 (0.08+0.24) | 32.08 |
| + LoFiT | 10% | 10% | 0.34 (0.15+0.19) | 74.71 | 0.40 (0.21+0.19) | 41.97 | 0.27 (0.08+0.19) | 31.04 |
| + *P2D‡ (Ours)* | 10% | 10% | 0.32 (0.15+0.17) | **78.82** | 0.38 (0.21+0.17) | **44.04** | 0.25 (0.08+0.17) | **32.74** |

*Table 24.* Evaluation results for various data selection and fine-tuning strategies on GSM8K, DialogSum, and BioInstruct using Qwen-2.5-7B-Instruct (Q2.5-7), Qwen-3-8B (Q3-8), and Llama-3-8B-Instruct (L3-8). All comparison methods we employ here constrain data selection to 30% of examples and fine-tuning to updating only 30% of attention heads. P2D† denotes our proposed Parameter-Guided Data Selection, while P2D‡ represents our Sparse Head Adaptation. The best results are highlighted in **bold**.

| Method | $\rho_D$ | $\rho_P$ | GSM8K | | | DialogSum | | | BioInstruct | | | Avg. |
|---|---|---|---|---|---|---|---|---|---|---|---|---|
| | | | Q2.5–7 | Q3–8 | L3–8 | Q2.5–7 | Q3–8 | L3–8 | Q2.5–7 | Q3–8 | L3–8 | |
| *Vanilla* | – | – | 62.31 | 65.73 | 48.62 | 25.41 | 22.97 | 20.16 | 16.35 | 12.94 | 12.92 | *31.93* |
| *Full (full dataset)* | | | | | | | | | | | | |
| + full parameters | 100% | 100% | 81.55 | 84.89 | 68.63 | 48.36 | 44.22 | 46.89 | 38.01 | 42.16 | 37.59 | 54.70 |
| + random | 100% | 30% | 69.12 | 71.56 | 59.23 | 42.15 | 40.12 | 41.34 | 33.45 | 39.12 | 30.89 | 47.44 |
| + LoRA | 100% | 30% | 80.85 | 83.12 | 67.95 | 49.56 | 44.65 | 46.89 | 38.89 | **42.25** | **38.65** | 54.75 |
| + LoFiT | 100% | 30% | 80.03 | 82.28 | 67.41 | 48.79 | 43.92 | 45.83 | 37.54 | 41.07 | 37.91 | 53.86 |
| *+ P2D‡ (Ours)* | 100% | 30% | **82.95** | **85.88** | **69.45** | **50.15** | **45.92** | **47.88** | **40.56** | 40.35 | 38.12 | **55.70** |
| *IFD* | | | | | | | | | | | | |
| + full parameters | 30% | 100% | 74.56 | 76.23 | 55.45 | 36.12 | 33.56 | 32.89 | 25.12 | **26.45** | 24.56 | 42.77 |
| + random | 30% | 30% | 68.45 | 68.12 | 53.89 | 33.89 | 32.45 | 31.56 | 24.45 | 23.89 | 22.98 | 39.96 |
| + LoRA | 30% | 30% | 76.12 | 77.05 | 56.89 | 36.89 | 34.78 | 33.12 | **25.89** | 26.12 | **25.45** | 43.59 |
| + LoFiT | 30% | 30% | 75.33 | 76.28 | 55.72 | 35.94 | 33.67 | 32.09 | 25.13 | 25.29 | 24.81 | 42.70 |
| *+ P2D‡ (Ours)* | 30% | 30% | **77.25** | **78.15** | **57.65** | **37.56** | **35.45** | **33.95** | 25.65 | 26.05 | 25.15 | **44.10** |
| *Nuggets* | | | | | | | | | | | | |
| + full parameters | 30% | 100% | 72.45 | 75.12 | 54.56 | 35.12 | 31.12 | 34.89 | 26.56 | 25.67 | 26.89 | 42.49 |
| + random | 30% | 30% | 66.89 | 68.95 | 54.45 | 32.56 | 31.05 | 33.89 | 23.89 | 25.12 | 24.45 | 40.14 |
| + LoRA | 30% | 30% | 74.12 | 76.05 | 54.89 | 35.65 | **34.89** | 37.12 | 29.56 | **28.65** | 27.89 | 44.31 |
| + LoFiT | 30% | 30% | 73.08 | 75.29 | 53.84 | 34.72 | 34.03 | 36.19 | 28.47 | 27.53 | 27.11 | 43.36 |
| *+ P2D‡ (Ours)* | 30% | 30% | **75.35** | **76.95** | **56.12** | **36.45** | 34.56 | **37.56** | **29.89** | 28.45 | **28.15** | **44.83** |
| *Data Whisperer* | | | | | | | | | | | | |
| + full parameters | 30% | 100% | 75.56 | 78.45 | 62.56 | 45.67 | 43.12 | 42.15 | 30.89 | 29.56 | 28.45 | 48.49 |
| + random | 30% | 30% | 68.12 | 70.15 | 57.65 | 39.45 | 37.12 | 35.56 | 26.45 | 25.15 | 24.35 | 42.67 |
| + LoRA | 30% | 30% | 77.15 | 79.89 | 63.65 | 44.56 | 42.12 | 40.56 | 29.65 | 27.89 | 27.15 | 48.07 |
| + LoFiT | 30% | 30% | 76.24 | 79.03 | 62.58 | 43.39 | 41.06 | 39.77 | 28.84 | 26.92 | 26.18 | 47.11 |
| *+ P2D‡ (Ours)* | 30% | 30% | **78.56** | **80.95** | **65.15** | **45.56** | **42.65** | **41.45** | **30.65** | **30.25** | **28.89** | **49.35** |
| *P2D† (Ours)* | | | | | | | | | | | | |
| + full parameters | 30% | 100% | 80.20 | 82.56 | 67.12 | 46.12 | 43.65 | 42.25 | 34.12 | 34.56 | 33.45 | 51.56 |
| + random | 30% | 30% | 71.56 | 70.12 | 57.12 | 40.89 | 39.56 | 38.89 | 31.45 | 29.65 | 30.15 | 45.60 |
| + LoRA | 30% | 30% | 80.45 | 81.65 | 66.89 | 46.15 | 43.15 | 42.56 | 35.65 | **36.12** | 33.89 | 51.83 |
| + LoFiT | 30% | 30% | 79.62 | 80.84 | 66.17 | 45.39 | 42.23 | 41.87 | 34.72 | 35.06 | 33.14 | 51.00 |
| *+ P2D‡ (Ours)* | 30% | 30% | **81.95** | **83.15** | **67.45** | **47.12** | **44.15** | **43.56** | **36.85** | 35.89 | **34.95** | **52.79** |

*Table 25.* Evaluation results for various data selection and fine-tuning strategies on GSM8K, DialogSum, and BioInstruct using Qwen-2.5-7B-Instruct (Q2.5-7), Qwen-3-8B (Q3-8), and Llama-3-8B-Instruct (L3-8). All comparison methods we employ here constrain data selection to 50% of examples and fine-tuning to updating only 50% of attention heads. P2D$^\dagger$ denotes our proposed Parameter-Guided Data Selection, while P2D$^\ddagger$ represents our Sparse Head Adaptation. The best results are highlighted in **bold**.

| Method | $\rho_D$ | $\rho_P$ | GSM8K | | | DialogSum | | | BioInstruct | | | Avg. |
|---|---|---|---|---|---|---|---|---|---|---|---|---|
| | | | Q2.5–7 | Q3–8 | L3–8 | Q2.5–7 | Q3–8 | L3–8 | Q2.5–7 | Q3–8 | L3–8 | |
| ***Full (full dataset)*** | | | | | | | | | | | | |
| + full parameters | 100% | 100% | 81.62 | 84.93 | 68.51 | 48.41 | 44.18 | 46.94 | 37.96 | 42.21 | 37.64 | 54.71 |
| + random | 100% | 50% | 71.34 | 73.08 | 60.82 | 43.91 | 41.47 | 43.05 | 34.38 | 39.13 | 31.52 | 48.74 |
| + LoRA | 100% | 50% | 81.13 | 83.21 | 68.17 | 49.58 | 45.06 | 47.41 | 39.23 | **42.34** | **38.87** | 55.00 |
| + LoFiT | 100% | 50% | 80.39 | 82.54 | 67.43 | 48.87 | 44.29 | 46.62 | 38.61 | 41.28 | 38.05 | 54.23 |
| + *P2D$^\ddagger$ (Ours)* | 100% | 50% | **82.51** | **85.37** | **69.42** | **50.18** | **46.13** | **48.31** | **40.62** | 41.58 | 38.41 | **55.84** |
| ***IFD*** | | | | | | | | | | | | |
| + full parameters | 50% | 100% | 74.91 | 76.34 | 57.23 | 36.82 | 34.39 | 33.56 | 24.88 | 25.84 | 24.16 | 43.13 |
| + random | 50% | 50% | 68.42 | 69.93 | 55.17 | 33.81 | 31.36 | 30.92 | 24.61 | 23.82 | 22.89 | 40.10 |
| + LoRA | 50% | 50% | 76.03 | 77.41 | 58.16 | 36.88 | 34.72 | 32.91 | 25.53 | 26.04 | 24.92 | 43.62 |
| + LoFiT | 50% | 50% | 75.21 | 76.68 | 57.34 | 36.09 | 33.84 | 32.05 | 25.07 | 25.51 | 24.38 | 42.91 |
| + *P2D$^\ddagger$ (Ours)* | 50% | 50% | **77.24** | **78.53** | **59.41** | **37.92** | **35.61** | **33.88** | **26.58** | **26.91** | **25.68** | **44.64** |
| ***Nuggets*** | | | | | | | | | | | | |
| + full parameters | 50% | 100% | 72.41 | 75.09 | 55.08 | **43.07** | 39.82 | **40.79** | 28.16 | 27.13 | **27.84** | 45.49 |
| + random | 50% | 50% | 67.62 | 69.04 | 54.91 | 37.13 | 34.82 | 36.15 | 24.64 | 23.71 | 23.49 | 41.28 |
| + LoRA | 50% | 50% | 74.18 | 76.13 | 56.27 | 40.72 | 38.61 | 39.94 | 27.91 | **27.08** | 26.53 | 45.26 |
| + LoFiT | 50% | 50% | 73.45 | 75.38 | 55.42 | 39.87 | 37.93 | 39.18 | 27.34 | 26.59 | 25.87 | 44.56 |
| + *P2D$^\ddagger$ (Ours)* | 50% | 50% | **75.61** | **77.28** | **57.39** | 41.48 | **39.92** | 40.51 | **28.71** | 26.98 | 27.19 | **46.12** |
| ***Data Whisperer*** | | | | | | | | | | | | |
| + full parameters | 50% | 100% | 75.39 | 78.04 | 64.12 | 45.08 | 42.29 | 41.83 | 29.91 | 28.27 | 27.36 | 48.03 |
| + random | 50% | 50% | 68.37 | 70.31 | 58.09 | 39.18 | 37.09 | 35.52 | 25.68 | 24.41 | 23.84 | 42.50 |
| + LoRA | 50% | 50% | 77.24 | 80.08 | 63.82 | 44.63 | 41.91 | 40.58 | 28.53 | 26.77 | 26.14 | 47.74 |
| + LoFiT | 50% | 50% | 76.42 | 79.27 | 63.14 | 43.89 | 41.13 | 39.71 | 27.91 | 26.18 | 25.49 | 47.02 |
| + *P2D$^\ddagger$ (Ours)* | 50% | 50% | **78.71** | **81.18** | **65.23** | **45.49** | **43.16** | **41.88** | **30.19** | **28.87** | **27.48** | **49.13** |
| ***P2D$^\dagger$ (Ours)*** | | | | | | | | | | | | |
| + full parameters | 50% | 100% | 81.28 | 83.87 | 68.51 | 45.94 | 43.46 | 42.12 | 31.34 | 33.88 | 31.79 | 51.35 |
| + random | 50% | 50% | 74.12 | 72.39 | 59.28 | 40.09 | 38.64 | 38.17 | 26.88 | 24.91 | 25.18 | 44.41 |
| + LoRA | 50% | 50% | 80.64 | 82.09 | 69.13 | 45.17 | 42.53 | 41.52 | 31.02 | 34.03 | 31.18 | 50.81 |
| + LoFiT | 50% | 50% | 79.91 | 81.24 | 68.17 | 44.48 | 41.79 | 40.81 | 30.22 | 33.31 | 30.49 | 50.05 |
| + *P2D$^\ddagger$ (Ours)* | 50% | 50% | **82.31** | **83.39** | **69.91** | **46.62** | **44.29** | **43.18** | **32.49** | **35.11** | **32.61** | **52.21** |

*Table 26.* Evaluation results for various data selection and fine-tuning strategies on GSM8K, DialogSum, and BioInstruct using Qwen-2.5-7B-Instruct (Q2.5-7), Qwen-3-8B (Q3-8), and Llama-3-8B-Instruct (L3-8). All comparison methods we employ here constrain data selection to 70% of examples and fine-tuning to updating only 70% of attention heads. P2D$^\dagger$ denotes our proposed Parameter-Guided Data Selection, while P2D$^\ddagger$ represents our Sparse Head Adaptation. The best results are highlighted in **bold**.

| Method | $\rho_D$ | $\rho_P$ | GSM8K | | | DialogSum | | | BioInstruct | | | Avg. |
|---|---|---|---|---|---|---|---|---|---|---|---|---|
| | | | Q2.5–7 | Q3–8 | L3–8 | Q2.5–7 | Q3–8 | L3–8 | Q2.5–7 | Q3–8 | L3–8 | |
| ***Full (full dataset)*** | | | | | | | | | | | | |
| + full parameters | 100% | 100% | 81.63 | 84.82 | 68.59 | 48.42 | 44.15 | 46.91 | 38.08 | 42.19 | 37.64 | 54.71 |
| + random | 100% | 70% | 69.87 | 71.53 | 59.68 | 43.12 | 40.71 | 42.19 | 34.61 | 40.08 | 31.72 | 48.17 |
| + LoRA | 100% | 70% | 80.59 | 83.14 | 67.52 | 50.18 | 45.27 | 47.41 | 39.83 | **43.09** | **39.31** | 55.15 |
| + LoFiT | 100% | 70% | 79.84 | 82.36 | 66.91 | 49.32 | 44.58 | 46.77 | 38.96 | 42.14 | 38.53 | 54.38 |
| *+ P2D$^\ddagger$ (Ours)* | 100% | 70% | **81.98** | **84.71** | **69.18** | **50.63** | **46.49** | **48.47** | **41.29** | 42.18 | 38.79 | **55.97** |
| ***IFD*** | | | | | | | | | | | | |
| + full parameters | 70% | 100% | 75.96 | 77.92 | 58.09 | 38.16 | 36.21 | 35.13 | 27.21 | **29.13** | 27.42 | 45.03 |
| + random | 70% | 70% | 69.24 | 70.51 | 55.28 | 33.19 | 31.28 | 30.73 | 24.84 | 23.98 | 23.14 | 40.24 |
| + LoRA | 70% | 70% | 75.18 | 76.39 | 57.11 | 36.38 | 33.81 | 32.72 | 26.54 | 25.68 | 24.98 | 43.20 |
| + LoFiT | 70% | 70% | 74.32 | 75.56 | 56.47 | 35.82 | 32.94 | 31.89 | 25.91 | 24.87 | 24.23 | 42.45 |
| *+ P2D$^\ddagger$ (Ours)* | 70% | 70% | **76.49** | **77.89** | **58.39** | **37.51** | **34.89** | **33.98** | **27.61** | 26.89 | **25.89** | **44.39** |
| ***Nuggets*** | | | | | | | | | | | | |
| + full parameters | 70% | 100% | **73.57** | **76.51** | 55.98 | **36.41** | 32.28 | 36.12 | 27.86 | 26.63 | **28.21** | 43.73 |
| + random | 70% | 70% | 67.16 | 68.42 | 57.06 | 32.44 | 30.89 | 32.34 | 24.25 | 23.84 | 23.24 | 39.96 |
| + LoRA | 70% | 70% | 73.09 | 75.83 | 56.17 | 34.49 | **33.82** | 35.97 | **28.49** | **27.39** | 26.16 | 43.49 |
| + LoFiT | 70% | 70% | 72.31 | 74.96 | 55.42 | 33.86 | 33.14 | 35.28 | 27.88 | 26.74 | 25.59 | 42.80 |
| *+ P2D$^\ddagger$ (Ours)* | 70% | 70% | 73.49 | 76.19 | **57.29** | 35.69 | 33.19 | **36.49** | 27.99 | 27.19 | 27.49 | **43.89** |
| ***Data Whisperer*** | | | | | | | | | | | | |
| + full parameters | 70% | 100% | 78.58 | 80.93 | 65.77 | 45.21 | 42.87 | 41.99 | **30.62** | **29.18** | 28.07 | 49.25 |
| + random | 70% | 70% | 70.19 | 71.83 | 59.02 | 38.99 | 36.71 | 35.49 | 25.09 | 24.14 | 23.91 | 42.82 |
| + LoRA | 70% | 70% | 78.68 | 81.24 | 66.13 | 44.21 | 41.62 | 40.31 | 29.21 | 27.53 | 26.86 | 48.42 |
| + LoFiT | 70% | 70% | 77.94 | 80.46 | 65.28 | 43.56 | 40.89 | 39.54 | 28.67 | 26.81 | 26.15 | 47.70 |
| *+ P2D$^\ddagger$ (Ours)* | 70% | 70% | **79.89** | **82.41** | **67.31** | **45.49** | **43.19** | **42.09** | 30.19 | 28.89 | **28.19** | **49.74** |
| ***P2D$^\dagger$ (Ours)*** | | | | | | | | | | | | |
| + full parameters | 70% | 100% | 81.53 | 84.18 | 68.89 | 45.92 | 43.51 | 42.08 | 31.32 | 33.91 | 31.83 | 51.46 |
| + random | 70% | 70% | 74.58 | 72.91 | 59.18 | 39.79 | 38.28 | 37.99 | 26.53 | 24.61 | 25.21 | 44.34 |
| + LoRA | 70% | 70% | 81.29 | 82.47 | 69.62 | 45.06 | 42.17 | 41.49 | 30.93 | 34.06 | 30.68 | 50.86 |
| + LoFiT | 70% | 70% | 80.56 | 81.74 | 68.83 | 44.29 | 41.52 | 40.81 | 30.14 | 33.28 | 29.96 | 50.13 |
| *+ P2D$^\ddagger$ (Ours)* | 70% | 70% | **83.89** | **86.44** | **71.13** | **50.15** | **45.72** | **47.76** | **40.85** | **40.26** | **38.37** | **56.06** |

