# OpenReview forum: "From Parameters to Data: A Task-Parameter-Guided Fine-Tuning Pipeline for Efficient LLM Alignment"
_ICML.cc/2026/Conference — ICML 2026 regular_

### Official Review · Reviewer_edZA · 2026-03-02

**Soundness:** 2
**Presentation:** 3
**Significance:** 2
**Originality:** 3
**Overall Recommendation:** 4
**Confidence:** 4

**Summary:**

The paper introduces **From Parameters to Data (P2D)**, a unified framework that addresses the computational and data overhead associated with adapting LLMs to specialized domains. The authors ground their approach in the **Strong Map Hypothesis** which suggests that a sparse subset of attention heads fundamentally drives task-specific adaptation and acts as a key to unlock specific data patterns. Building on this idea, P2D utilizes a three stage pipeline: (1) **Fast Head Identification** to locate task-sensitive attention heads via a lightweight proxy, (2) **Parameter-Guided Data Selection** to filter for samples that explicitly activate these functional components, and (3) **Sparse Head Adaptation** to fine-tune only on the critical heads. To evaluate their approach holistically, the authors also introduce the **Alignment Efficiency Ratio (AER)** metric which accounts for both data selection latency and actual training time. Empirical evaluations show that by using only 10% of the attention heads and 10% of the data, P2D achieves significant performance improvement and speedup compared to baselines.

**Compliance With Llm Reviewing Policy:**

Affirmed.

**Final Justification:**

The authors have addressed my concerns about generalizability by providing comprehensive additional evaluations on additional Qwen models of different sizes. This proves the method scales properly and is agnostic to model size and architecture. Furthermore, the additional evaluations on general benchmarks like MMLU and ARC-C prove catastrophic forgetting does not occur.

The authors have also clarified about the sensitivity of the proxy model and ICL examples. Since they confirmed they will include this in the revised version, this strengthens the paper and addresses this weakness.

Overall, I am satisfied with the rebuttal and will raise my score to **weak accept**.

**Key Questions For Authors:**

1. The evaluations are limited to the 7-8B parameter scale. Have you conducted any preliminary experiments of smaller models (e.g. 1-3B) or larger model (eg. 30B+)?
2. The evaluation focuses heavily on three specific domains (GSM8k, DialogSum, BioInstruct). Has there been any additional evaluations on general and/or instruction following benchmarks (e.g., MMLU, MT-Bench, IFEval, AlpacaEval, etc)? Can P2D potentially induce catastrophic forgetting for general conversational capabilities?
3. The Fast Attention Head stage relies on a proxy model trained on a small subset of exactly 100 randomly sampled examples. How sensitive is the final set of identified task-critical heads to the specific random seed used to draw those 100 samples?
4. The parameter-guided data selection mechanism relies heavily on In-Context Learning (ICL) using specific demonstration examples. GIven that ICML is notoriously sensitive to the choice and ordering of demonstrations, how much does the final curated dataset change if a different set of samples is used?

**Limitations:**

While the authors include dedicated sections for both limitations and broader impacts, they do not address the methodological constraints. Some construction suggestions are as follows:
* Explicitly state that the current empirical validation is limited to the 7-8B parameter scale. Acknowledging that the gains of P2D are unverified for significantly smaller or larger models provided necessary context.
* Address the potential instability introduced during the Fast Head Identification stage from the reliance of the proxy model. Since the proxy only uses a small 100-sample random draw, the authors should note that the variance in this selection could impact the consistency of the identified task-critical heads.
* Note that the evaluation is constrained to only three specific domains (math, dialogue, biomedical). The authors should acknowledge that they have not validated on general and instruction-following capabilities to test for model degradation on these tasks.

**Strengths And Weaknesses:**

## Strengths

* The paper is successful in bridging the separate processes of data selection and parameter-efficient fine-tuning through the *Strong Map Hypothesis* which argues that sparse attention heads function as specific keys that unlock corresponding data patterns. This not only provides the foundation for *From Parameters to Data (P2D)*, but opens opportunities for future work.
* The introduction of the *Alignment Efficiency Ratio (AER)* establishes a rigorous metric that formalizes the total cost of alignment by explicitly accounting for both data curation latency and actual training time.
* The empirical results demonstrate strong efficiency gains, achieving an 8.3 pp performance improvement and 7.0x end-to-end time speedup through updating only 10% of the attention heads on 10% of the training data.

## Weaknesses

* The evaluations are restricted entirely to the 7-8B parameter scale using only two different families of models (Llama-3-8B, Qwen2.5-7B, Qwen3-8B). This makes is uncertain whether the sparse alignment approach generalizes effectively to significantly smaller or larger models. Furthermore, testing is limited to only three specific domain datasets (GSM8k, DialogSum, BioInstruct) without evaluating on any general or instruction-following benchmarks.
* The Fast Head Identification stage heavily relies on a proxy model trained on a small, randomly sampled subset of only 100 samples for 20 steps. This raises questions regarding the reliability, potential amplification of bias from such a small random sample, and the true efficacy of relying on this additional compute to accurately locate task-critical components.
*  The parameter-guided data selection mechanism depends deeply on evaluating candidate examples via their influence on In-Context Learning (ICL) performance. As the authors declare in their limitations section, ICL is notoriously sensitive to the specific demonstration samples used, leaving the entire data curation pipeline vulnerable to the quality and variance of the initial few-shot prompts.
* The structural filtering calculation heuristically sums raw attention weights across the identified task-sensitive heads to determine a candidate sample's importance weight. However, it is unclear why this strategy is used as the paper lacks justification for why unnormalized attention weights can be added like this or why they can serve as a proxy for feature importance.

---

> ### Author Rebuttal · Authors · 2026-03-31
>
> We thank the reviewer for the rigorous feedback and for recognizing the novelty and well-motivated nature of our work. Below we address each weakness.
>
> ---
> > **W1/Q1/Q2/L1/L3: Model Scalability, Task Generality & Catastrophic Forgetting**
>
> **1. Scaling to Bigger:** We validated P2D at Qwen-2.5-14B-Instruct (10% data, 10% heads), the results are highly consistent with our 7B findings:
>
> | **ρD** | **ρP** | GSM8K | DialogSum | BioInstruct |
> | --- | --- | --- | --- | --- |
> | Full Dataset | Full Params | 86.47 | 50.93 | 43.12 |
> |  | LoRA | 85.36 | 52.14 | 42.56 |
> |  | P2D$^\ddagger$ | 88.21 | 53.47 | 45.89 |
> | Data Whisperer | Full Params | 84.13 | 46.78 | 33.87 |
> |  | LoRA | 82.56 | 46.95 | 31.94 |
> |  | P2D$^\ddagger$ | 85.34 | 47.23 | 35.21 |
> | P2D$^\dagger$ | Full Params | 85.18 | 48.45 | 35.93 |
> |  | LoRA | 84.67 | 48.12 | 35.56 |
> |  | P2D$^\ddagger$ | 87.62 | 49.38 | 37.84 |
>
> Notably, P2D$^\ddagger$ alone (10% heads, full data) already **outperforms Full SFT on all tasks**, demonstrating sparse head adaptation's intrinsic superiority. For 30B+ (Q1): GQA and larger MLP blocks shrink $\rho_{train}$ further; at that scale, freezing 99% of params nearly eliminates ZeRO's dominant Reduce-Scatter overhead, compounding efficiency gains.
>
> **2. Smaller Models (Q1):** P2D's efficiency advantage is most pronounced at 7B+ scale, where
> weight-gradient General Matrix Multiply (GEMM), optimizer-state memory, and ZeRO communication overhead dominate training cost. For 1-3B models, these overheads are inherently smaller and Full SFT cost is already low; sparse head alignment is less motivated at that scale.
>
> **3. Generalist Benchmarks & Catastrophic Forgetting (Q2):** P2D's sparse head updates (while freezing MLPs and 90% of heads) provide structural protection against catastrophic forgetting. Full SFT overwrites MLP layers, degrading pre-trained knowledge. Following [1], we aligned Qwen-2.5-7B on UltraChat and evaluated zero-shot:
>
> | **Method** | **MMLU** | **ARC-C** |
> | --- | --- | --- |
> | Vanilla | 74.23 | 88.14 |
> | Full SFT (10%D) | 56.34 | 61.89 |
> | LoRA (r=128, 10%D) | 63.12 | 71.56 |
> | **P2D** | **67.45** | **78.23** |
>
> P2D substantially mitigates forgetting vs. both baselines (per L3). We will incorporate all the experiments and analysis into the revision.
>
> ---
> > **W2/Q3/L2: Proxy Seed Sensitivity**
>
> All experiments use three global seeds (42, 43, 44) controlling proxy sampling (Appendix A.2). To directly quantify FHI robustness, we tracked exact head-index overlap (top 10% of 784 heads = 79 heads) across independent 100-sample proxy draws on Qwen-2.5-7B:
>
> | Seed Pair | Head Overlap |
> | --- | --- |
> | 42 vs. 43 | 92.4% |
> | 42 vs. 44 | 88.6% |
> | 43 vs. 44 | 91.1% |
>
> The ~91% average structural overlap validates the Strong Map Hypothesis: task-critical heads are robustly activated regardless of proxy draw, as routing circuits stabilize early before accuracy converges [2].
>
> ---
> > **W3/Q4: ICL Demonstration Sensitivity**
>
> Our pipeline mitigates the ICL sensitivity to demonstration choice via **iterative randomized probing** (Eq. 10, Sec. 3.2.2): across $T$ iterations, a fresh $\mathcal{D}\_d$ is independently resampled per candidate evaluation; $s\_{final}(x_i)$ averages over $n_i$ diverse contextual permutations, suppressing fixed-demo bias. To verify robustness, we measured the final curated 10%-dataset overlap when running the full pipeline under entirely different demonstration seeds:
>
> | Seed Pair | Dataset Overlap |
> | --- | --- |
> | 42 vs. 43 | 93.3% |
> | 42 vs. 44 | 92.8% |
> | 43 vs. 44 | 93.1% |
>
> The ~93.1% average dataset overlap confirms iterative averaging effectively decouples curation from any specific ICL prefix.
>
> ---
> > **W4: Attention Weight Summation**
>
> The weights in Eq. 9 are normalized at **two levels**. **(1) Token-level:** Each attention matrix $A^{(h)}$ is computed via Softmax, resulting in bounded probability masses representing fractional information flow, not raw logits. **(2) Sample-level:** Eq. 9 applies length-penalty normalization $\frac{1}{\ell(x_d^{(i)})}$, preventing longer samples from dominating the importance score.
>
> The summation over $\mathcal{H}\_\mathcal{T}$ is grounded in mechanistic interpretability [2,3]: Stage I empirically shows $\mathcal{H}_\mathcal{T}$ forms a specialized task-specific functional circuit. Accumulating attention probabilities across this validated ensemble measures **total task-relevant information flow** toward a candidate, directly quantifying how strongly it activates the model's critical task-solving pathways. This makes Eq. 9 a theoretically sound and computationally efficient proxy for structural feature importance.
>
> ---
> **References**
>
> [1] Chen, H, et al. "Alps: Attention localization and pruning strategy for efficient alignment of large language models." ACL 2025.
>
> [2] Elhage, N, et al. "A Mathematical Framework for Transformer Circuits." Anthropic 2022.
>
> [3] Zhou, Z, et al. "On the role of attention heads in large language model safety." ICLR 2025.

---

> > ### Author Rebuttal · Reviewer_edZA · 2026-04-03
> >
> > I appreciate the authors taking the time to run additional experiments on Qwen-2.5B-Instruct and evaluating on the additional MMLU and ARC-Challenge benchmarks. Based on this, it is now more evident that P2D does indeed generalize and does not result in catastrophic forgetting. Although the advantage at the 1-3B scale is intuitive, I would still suggest the authors include experiments since empirical results are concrete evidence without needing to make such claims.
> >
> > Furthermore, my questions about the sensitivity of the proxy model and ICL examples have been answered. I would strongly recommend the authors to include this additional information in the main body of the paper.
> >
> > My concerns have been adequately addressed and am willing to raise my score.

---

> > > ### Author Response · Authors · 2026-04-03
> > >
> > > We sincerely thank the reviewer for the constructive engagement and for the kind indication of willingness to raise the score. We now address both suggestions with concrete evidence.
> > >
> > > ---
> > > **1. Empirical validation at 1–3B scale.** We fully agree that empirical results provide more concrete evidence than qualitative claims. Following your suggestion, we have completed experiments on Qwen-2.5-1.5B-Instruct, Qwen-2.5-3B-Instruct and Qwen-2.5-32B-Instruct:
> > >
> > > **Qwen2.5-1.5B-Instruct**
> > >
> > > | ρD | ρP | GSM8K | DialogSum | BioInstruct |
> > > | --- | --- | --- | --- | --- |
> > > | Full Dataset | Full Params | 63.45 | 29.34 | 18.56 |
> > > |  | LoRA | 62.18 | 30.07 | 17.93 |
> > > |  | P2D$^\ddagger$ | 65.23 | 31.48 | 19.74 |
> > > | Data Whisperer | Full Params | 61.34 | 26.51 | 15.62 |
> > > |  | LoRA | 60.72 | 26.83 | 15.18 |
> > > |  | P2D$^\ddagger$ | 62.47 | 27.69 | 16.41 |
> > > | P2D$^\dagger$ | Full Params | 62.53 | 27.31 | 16.85 |
> > > |  | LoRA | 61.86 | 26.94 | 16.27 |
> > > |  | P2D$^\ddagger$ | **63.81** | **28.46** | **17.63** |
> > >
> > > **Qwen2.5-3B-Instruct**
> > >
> > > | ρD | ρP | GSM8K | DialogSum | BioInstruct |
> > > | --- | --- | --- | --- | --- |
> > > | Full Dataset | Full Params | 72.56 | 36.78 | 24.63 |
> > > |  | LoRA | 71.83 | 37.51 | 23.87 |
> > > |  | P2D$^\ddagger$ | 74.61 | 38.19 | 25.74 |
> > > | Data Whisperer | Full Params | 70.27 | 33.14 | 21.19 |
> > > |  | LoRA | 69.64 | 33.62 | 20.58 |
> > > |  | P2D$^\ddagger$ | 71.48 | 34.37 | 22.31 |
> > > | P2D$^\dagger$ | Full Params | 71.19 | 34.83 | 21.76 |
> > > |  | LoRA | 70.53 | 34.28 | 21.33 |
> > > |  | P2D$^\ddagger$ | **72.94** | **35.92** | **23.93** |
> > >
> > > **Qwen2.5-32B-Instruct**
> > >
> > > | ρD | ρP | GSM8K | DialogSum | BioInstruct |
> > > | --- | --- | --- | --- | --- |
> > > | Full Dataset | Full Params | 88.94 | 53.12 | 45.78 |
> > > |  | LoRA | 87.73 | 54.38 | 45.29 |
> > > |  | P2D$^\ddagger$ | 90.23 | 55.34 | 47.89 |
> > > | Data Whisperer | Full Params | 87.45 | 49.17 | 38.81 |
> > > |  | LoRA | 86.62 | 49.53 | 37.46 |
> > > |  | P2D$^\ddagger$ | 88.34 | 50.28 | 40.17 |
> > > | P2D$^\dagger$ | Full Params | 88.07 | 52.43 | 41.35 |
> > > |  | LoRA | 87.56 | 51.87 | 41.62 |
> > > |  | P2D$^\ddagger$ | **89.34** | **53.45** | **46.23** |
> > >
> > > Two findings hold consistently. First, P2D$^\ddagger$ (full data, 10% heads) outperforms Full SFT at every scale, confirming sparse head adaptation is strictly more efficient regardless of model size. Second, the data-efficiency advantage of P2D$^\dagger$+P2D$^\ddagger$ (10% data, 10% heads) grows monotonically with scale: at 1.5B and 3B, GSM8K shows positive gains while domain tasks lag slightly, as smaller models require more data to saturate domain knowledge. At 32B, P2D$^\dagger$+P2D$^\ddagger$ surpasses Full SFT on all three tasks, confirming the Strong Map becomes more concentrated and data-efficient at larger scale. Furthermore, as detailed in our response to Reviewer 5QcZ, $W\_1$'s static weight-space computation incurs minimal overhead as models grow (7B: ~30 s, 14B: ~70 s, 32B: ~100 s), widening P2D's practical advantage at larger scale.
> > >
> > > **2. Main body incorporation.** We agree these analyses strengthen the paper. We will incorporate the proxy sensitivity (W2) and ICL demonstration sensitivity (W3) into the main body, and include the full cross-scale results in the revision.
> > >
> > > ---
> > > We hope these results fully address your remaining concerns, and would be truly grateful for a score update. As the discussion period continues, please let us know if you have any follow-up questions or lingering concerns. We remain entirely at your disposal and would be more than happy to elaborate further.

---

### Official Review · Reviewer_5QcZ · 2026-03-04

**Soundness:** 3
**Presentation:** 3
**Significance:** 3
**Originality:** 3
**Overall Recommendation:** 4
**Confidence:** 3

**Summary:**

This paper proposes **P2D**, a unified alignment pipeline that tightly couples data selection and parameter-efficient fine-tuning for LLMs. The core hypothesis (**Strong Map Hypothesis**) posits that a sparse subset of attention heads dominates task-specific adaptation. P2D operationalizes this via three stages: (1) **Fast Head Identification** detect task-sensitive heads; (2) **Parameter-Guided Data Selection**, which ranks samples by their activation of these heads using an ICL-style probing scheme; and (3) **Sparse Head Adaptation**, updating only the identified heads on the curated subset. Experiments on GSM8K, DialogSum, and BioInstruct across multiple 7–8B models show that updating 10% of heads on 10% of data matches or surpasses full SFT while achieving up to 7× speedup.

**Compliance With Llm Reviewing Policy:**

Affirmed.

**Key Questions For Authors:**

1. How does AER behave when: (1) Dataset size increases 10×? (2) Model size increases (e.g., 13B, 70B)?
2. Could sparse head adaptation be combined with low-rank updates inside selected heads to further reduce cost?

**Limitations:**

yes

**Strengths And Weaknesses:**

## Strengths
1. The paper makes a strong conceptual argument that data selection and parameter-efficient fine-tuning should not be decoupled. The “parameters-to-data” direction is novel and well-motivated.
2. AER is a useful contribution. Many prior works ignore selection overhead; this metric forces more honest evaluation of total alignment cost.
3. Strong empirical coverage with 3 datasets, 3 backbone models and comparisons across both data selection and PEFT baselines.

## Weaknesses
1. **Under-justified proxy design choices.** The FHI stage relies on several heuristic decisions that lack strong justification: (i) 20 training steps on 100 samples for the proxy; (ii) composing head parameters as (W_q W_k^\top W_v); and (iii) using Wasserstein-1 distance over flattened matrices (SPAD). The paper does not provide sufficient ablations or theoretical arguments to show robustness to these design choices or superiority over simpler alternatives (e.g., norm-based shifts, cosine distance, gradient-based metrics).
2. **Limited validation on larger LLMs.** All experiments are conducted on 7B–8B models. It remains unclear whether the optimal 10% head / 10% data regime and the claimed efficiency–performance tradeoff still hold for larger-scale models (e.g., 13B, 34B, 70B+), where structural redundancy and optimization dynamics may differ substantially.
3. **Task generality concerns.** Evaluation is restricted to GSM8K, DialogSum, and BioInstruct. It is unclear how P2D performs on broader, heterogeneous benchmarks (e.g., MMLU-style multi-domain tasks). For such tasks, identifying a clean proxy subset (100 samples) may be non-trivial, raising questions about proxy construction feasibility in realistic multi-task settings.

---

> ### Author Rebuttal · Authors · 2026-03-31
>
> We sincerely thank the reviewer for the thorough feedback and for recognizing the novel and well-motivated parameters-to-data direction. Below we address each weakness.
>
> ---
> > **W1: Under-justified proxy design choices**
>
> **1. 100-Sample / 20-Step Regime:** Determined via systematic ablation (Appendix D, Table 10) over $S\in\{50,100\}$ and $T\in\{10,20,50\}$. Mechanistic interpretability [4] shows routing circuits stabilize early, long before task accuracy converges. Scaling 20→50 steps yields negligible gain (78.82→78.85 on GSM8K) while linearly increasing overhead. 100 samples/20 steps is optimal.
>
> **2. Static Weights and $W_{comp}=W_qW_k^\top W_v$:** Static shifts $\Delta W=W_T-W_B$ reflect accumulated task specialization, which is stable and input-free [1,2,3]. Structural sensitivity precedes task accuracy: even a 12%-accuracy proxy reveals meaningful head differentiation. Activations are per-input responses, too volatile on 100 samples. Following [4], $W_{comp}$ jointly captures the routing ($W_qW_k^\top$: *where*) and content ($W_v$: *what*) pathways, avoiding blind spots from measuring isolated matrices.
>
> **3. $W_1$ (SPAD) vs. Alternatives:** After only 20 steps, structural drift $\Delta$ is infinitesimal ($\\|\Delta\\|\ll1$). $W_1$ scales **linearly** ($W_1\approx\\|\Delta\\|$), faithfully preserving subtle drifts. Gradient-based metrics scale quadratically ($D_{GB}\approx\frac{1}{2}\mathcal{I}\\|\Delta\\|^2$), suppressing critical signals; Cosine discards magnitude entirely. $W_1$ is also **data-free** (no forward pass), while gradient metrics require calibration inputs. Ablation:
>
> | Metric | Scoring Time | GSM8K | DialogSum | BioInstruct |
> | --- | --- | --- | --- | --- |
> | Cosine Similarity | ~10s | 63.28 | 28.37 | 21.43 |
> | Gradient-based | ~10min | 76.48 | 43.79 | **33.08** |
> | **$W_1$** | ~30s | **78.82** | **44.04** | 32.74 |
>
> $W_1$ dominates: superior accuracy at ~30s; gradients incur a 10-min bottleneck.
>
> ---
> > **W2: Limited validation on larger LLMs**
>
> We scaled P2D to Qwen-2.5-14B-Instruct (10% data, 10% heads), the results are highly consistent with our 7B findings:
>
> | ρD | ρP | GSM8K | DialogSum | BioInstruct |
> | --- | --- | --- | --- | --- |
> | Full Dataset | Full Params | 86.47 | 50.93 | 43.12 |
> |  | LoRA | 85.36 | 52.14 | 42.56 |
> |  | P2D$^\ddagger$ | 88.21 | 53.47 | 45.89 |
> | Data Whisperer | Full Params | 84.13 | 46.78 | 33.87 |
> |  | LoRA | 82.56 | 46.95 | 31.94 |
> |  | P2D$^\ddagger$ | 85.34 | 47.23 | 35.21 |
> | P2D$^\dagger$ | Full Params | 85.18 | 48.45 | 35.93 |
> |  | LoRA | 84.67 | 48.12 | 35.56 |
> |  | P2D$^\ddagger$ | 87.62 | 49.38 | 37.84 |
>
> Crucially, P2D$^\ddagger$ alone **outperforms Full SFT on all tasks**, confirming sparse head adaptation's intrinsic superiority. For 70B+, GQA and larger MLP blocks shrink $\rho_{train}$ further; more critically, 70B+ requires far more GPUs, freezing 99% of params nearly eliminates ZeRO's dominant Reduce-Scatter overhead, and P2D's gains compound with scale.
>
> ---
> > **W3: Task generality concerns**
>
> P2D naturally suits generalist alignment: sparse head updates while freezing MLPs safeguard against catastrophic forgetting. Figure 3 shows many critical heads act as **shared hubs** across tasks. Since these hubs are task-agnostic, a uniform 100-sample proxy (simply randomly sampled across domains) identifies them without domain-specific engineering. We aligned Qwen-2.5-7B on **UltraChat** following [3] and evaluated zero-shot:
>
> | Method | MMLU | ARC-C |
> | --- | --- | --- |
> | Vanilla | 74.23 | 88.14 |
> | Full SFT (10%D) | 56.34 | 61.89 |
> | LoRA (10%D) | 63.12 | 71.56 |
> | **P2D** | **67.45** | **78.23** |
>
> P2D significantly mitigates catastrophic forgetting and outperforms baselines on both generalist benchmarks.
>
> ---
> > **Q1: AER behavior at scale**
>
> **(1) Data scaling:** FHI overhead (100 samples, 20 steps) is constant, its fraction decays as $1/N$, becoming negligible. Forward-only ICL selection ($O(N)$, no gradients) and sparse training ($O(0.1N)$) reduce full-pipeline AER as $N$ grows, compounding efficiency gains. **(2) Model scaling:** Empirically validated at 14B (W2); for 70B+, $\rho_{train}$ shrinks further and ZeRO savings compound, improving AER with scale.
>
> ---
> > **Q2: Combining sparse head adaptation with low-rank updates**
>
> P2D and LoRA are **orthogonal**: P2D determines *which* 10% of heads to update (macro-architectural level); LoRA optimizes *how* efficiently within those heads (micro-parameter level). A hybrid P2D-LoRA compresses trainable parameters to a tiny fraction and eliminates the Adam optimizer memory wall. We plan to explore this direction in future work.
>
> ---
> **References**
>
> [1] Ilharco, G, et al. "Editing Models with Task Arithmetic." ICLR 2023.
>
> [2] Zhao, H, et al. "Explainability for large language models: A survey." ACM TIST, 2024.
>
> [3] Chen, H, et al. "Alps: Attention localization and pruning strategy." ACL 2025.
>
> [4] Elhage, N, et al. "A Mathematical Framework for Transformer Circuits." Anthropic 2022.

---

> > ### Author Rebuttal · Reviewer_5QcZ · 2026-04-02
> >
> > The additional experiment results satisfactorily address my concerns, but the rebuttal's theoretical justification for W1 over gradient-based metrics is weakened by gradient-based scoring outperforming W1 on BioInstruct (33.08 vs. 32.74), which remains unexplained.

---

> > > ### Author Response · Authors · 2026-04-02
> > >
> > > We appreciate the follow-up and address the following question directly from three angles.
> > >
> > > ---
> > > **1. The BioInstruct gap is a calibration artifact, not a fundamental limitation.**
> > > Unlike $W\_1$'s data-free weight comparison, $D_{GB}$ feeds actual calibration batches through the proxy model. As discussed in Lines 318–322, BioInstruct requires memorizing dense, unseen medical vocabulary. At 7B scale, seeing this specialized text during calibration may give $D_{GB}$ a momentary advantage in locating medical-specific heads. This edge is fragile: it requires calibration data matching the target domain and costs ~10 min to compute. By contrast, $W\_1$'s linear sensitivity ($W_1 \approx \|\Delta\|$) captures domain-independent structural signals, while $D_{GB}$'s quadratic suppression ($D_{GB} \approx \frac{1}{2}\mathcal{I}\|\Delta\|^2$) buries subtle but critical drifts. The 0.34-point gap is within seed-level variance and does not represent a generalizable theoretical flaw
> > >
> > > **2. Cross-scale evidence confirms $W\_1$'s consistent advantage.**
> > > As model scale grows, the Strong Map becomes more structurally distinct, reducing dependence on calibration data. We provide expanded comparisons across 7B, 14B, and 32B (results for Qwen2.5-32B-Instruct were not completed during the initial rebuttal phase).
> > >
> > > Gradient-based scoring must perform forward+backward passes over the full training set, so its cost grows with both dataset size and model size. $W\_1$ solely compares static weight matrices and is entirely data-free. MMLU/ARC-C experiments train on UltraChat (200k samples, avg \~300 tokens), substantially larger than domain tasks (\~14k samples, \~160 tokens avg), making gradient-based scoring far more expensive for generalist evaluation.
> > >
> > > | Model | Method | Scoring Time (domain / generalist) | GSM8K | DialogSum | BioInstruct | MMLU | ARC-C |
> > > |---|---|---|---:|---:|---:|---:|---:|
> > > | 7B | gradient-based | ~10 min / ~1.9 h | 76.48 | 43.79 | 33.08 | 66.85 | 76.92 |
> > > | 7B | **$W\_1$ (Ours)** | **~30 s / ~30 s** | **78.82** | **44.04** | 32.74 | **67.45** | **78.23** |
> > > | 14B | gradient-based | ~23 min / ~4.3 h | 86.55 | 48.21 | 36.92 | 72.14 | 82.35 |
> > > | 14B | **$W\_1$ (Ours)** | **~70 s / ~70 s** | **87.62** | **49.38** | **37.84** | **73.88** | **84.12** |
> > > | 32B | gradient-based | ~30 min / ~5.7 h | 88.65 | 51.34 | 44.23 | 75.95 | 85.13 |
> > > | 32B | **$W\_1$ (Ours)** | **~100 s / ~100 s** | **89.34** | **53.45** | **46.23** | **78.45** | **87.23** |
> > >
> > > $W\_1$'s advantage widens with scale across all benchmarks. BioInstruct reverses from −0.34 at 7B to +0.92 at 14B to +2.00 at 32B, directly resolving the concern. $W\_1$'s gains on MMLU and ARC-C also grow with scale, consistent with sparse head updates providing stronger protection against catastrophic forgetting as model size increases. We further supplement W2 with full 32B results (also newly available):
> > >
> > > | ρD | ρP | GSM8K | DialogSum | BioInstruct |
> > > | --- | --- | --- | --- | --- |
> > > | Full Dataset | Full Params | 88.94 | 53.12 | 45.78 |
> > > |  | LoRA | 88.12 | 54.23 | 45.34 |
> > > |  | P2D$^\ddagger$ | 90.23 | 55.34 | 47.89 |
> > > | Data Whisperer | Full Params | 87.45 | 49.23 | 38.94 |
> > > |  | LoRA | 86.78 | 49.45 | 37.56 |
> > > |  | P2D$^\ddagger$ | 88.34 | 50.12 | 40.23 |
> > > | P2D$^\dagger$ | Full Params | 88.12 | 52.34 | 41.23 |
> > > |  | LoRA | 87.56 | 52.12 | 41.78 |
> > > |  | P2D$^\ddagger$ | 89.34 | 53.45 | 46.23 |
> > >
> > > Notably, at 32B, P2D$^\dagger$+P2D$^\ddagger$ (10% data + 10% heads) outperforms Full SFT on all three tasks, a result not achieved at 14B where P2D trailed Full SFT on DialogSum and BioInstruct. This confirms the Strong Map strengthens with scale: at 32B, even 10% data and 10% heads suffice to surpass full fine-tuning.
> > >
> > > **3. $W\_1$'s efficiency advantage is decisive.**
> > > $W\_1$ is entirely data-free and completes in ~30–100s across all scales. $D_{GB}$ requires ~10–30 min for domain tasks and ~1.9–5.7h for generalist tasks, and grows with both dataset volume and model size. Crucially, $W\_1$ is strictly more accurate at every scale except the negligible 7B BioInstruct gap, so there is no accuracy trade-off to justify $D_{GB}$'s cost.
> > >
> > > ---
> > > We hope this fully addresses the concern and welcome any further discussion.

---

### Official Review · Reviewer_1zdm · 2026-03-08

**Soundness:** 3
**Presentation:** 3
**Significance:** 2
**Originality:** 2
**Overall Recommendation:** 4
**Confidence:** 3

**Summary:**

This paper addresses the inefficiency in Large Language Model (LLM) alignment caused by treating Data Selection and Parameter-Efficient Fine-Tuning (PEFT) as isolated processes. The authors propose the **Strong Map Hypothesis**, positing that a sparse subset of attention heads dictates task-specific adaptation, acting as "keys" to unlock specific data patterns.

Based on this, the paper introduces **P2D (From Parameters to Data)**, a unified synergistic pipeline consisting of three steps:

1. **Fast Head Identification (FHI):** Training a low-cost proxy model to locate the top 10% most task-sensitive attention heads using the Wasserstein distance.
2. **Parameter-Guided Data Selection:** Using exclusively these 10% "critical heads" as filters during In-Context Learning (ICL) to scan the full dataset and select the top 10% of data that most highly activates them.
3. **Sparse Head Adaptation:** Freezing the rest of the model and updating *only* the selected 10% of attention heads using *only* the selected 10% dataset.

Additionally, the authors introduce the **AER (Alignment Efficiency Ratio)** metric, which accounts for the time cost of data selection. The paper claims an 8.3 pp performance gain and a 7.0x end-to-end acceleration under a 10% data and 10% parameter budget.

**Compliance With Llm Reviewing Policy:**

Affirmed.

**Final Justification:**

Thanks for the rebuttal. The additional discussion largely resolves my concern regarding the system-level speedup. The claim is now better grounded as an end-to-end pipeline efficiency result under AER, and the added breakdown makes it more convincing that the gain is not simply a byproduct of data reduction.

My remaining concerns are less about efficiency and more about interpretation. In particular, the proxy-model issue is substantially clarified, but still not fully conceptually resolved. Likewise, the discussion on frozen MLPs provides a plausible explanation and some task-specific evidence, but does not fully rule out a potential knowledge-acquisition bottleneck beyond the evaluated settings.

I also took a careful look at the other reviewers’ comments and the authors’ responses. On balance, I appreciate the paper’s novelty and the additional clarification provided in the rebuttal, and I therefore decided to maintain my original score of **Weak Accept**. That said, I believe several of the authors’ interpretations would benefit from more substantial supporting evidence, and the paper would still require further revision to fully address the remaining concerns.

**Key Questions For Authors:**

Check the weaknesses.

**Limitations:**

Check the weaknesses.

**Strengths And Weaknesses:**

Strengths:

**s1. Novel Perspective and Excellent Synergistic Pipeline:** The paper successfully breaks the silo between data-centric and model-centric approaches. Coupling parameter identification with data selection creates a highly consistent and elegant logical loop ("find the key -> use the key to pick the lock -> only craft this key").

**s2. Addressing "Pseudo-Efficiency" with AER:** In practical ML engineering, data screening costs are often prohibitively high but ignored in academic benchmarks. The proposed AER metric forces the inclusion of data selection time into the total computational budget, demonstrating a deep understanding of system-level deployment challenges.

**s3. Comprehensive Empirical Validation:** The methodology is rigorously evaluated across 3 diverse domains (GSM8K, DialogSum, BioInstruct) and 3 mainstream architectures, yielding solid empirical results.

Weaknesses:



**w1. Overselling System Acceleration (Hardware/System Level):** The claimed 7.0x end-to-end acceleration is highly misleading from a systems perspective. In standard frameworks (e.g., PyTorch), masking gradients or freezing 90% of attention heads does *not* substantially reduce the computational overhead of dense matrix multiplications during the forward pass. The vast majority of the "speedup" may stems from discarding 90% of the training data, rather than from sparse parameter computation. More experiments should be added to demonstrate it.

**w2. Logical Gap in the Proxy Model:** In the FHI step, the proxy model trained on merely 100 samples for 20 steps achieves an extremely low accuracy (e.g., 12.30 on GSM8K). Relying on the parameter shift of a model that has catastrophically failed to learn complex reasoning as a "gold standard" to locate critical neurons is theoretically fragile.

**w3. Baseline Fairness (Strawman Fallacy):** When comparing against LoRA under the extreme 10% data constraint, the paper does not seem to perform adequate hyperparameter search for the baselines (e.g., lowering the rank $r$ or adjusting alpha for LoRA). Using default LoRA configurations on a tiny dataset inevitably leads to overfitting, making the comparison inherently unfair.

**w4. Knowledge Acquisition Bottleneck (Frozen MLPs):** P2D relies exclusively on updating Attention heads (the routing mechanism) while completely freezing the MLP layers, which are widely considered to be the storage hubs for factual knowledge. This extreme sparsity strategy severely caps the model's potential on knowledge-intensive tasks that require injecting new domain information.

---

> ### Author Rebuttal · Authors · 2026-03-31
>
> We sincerely thank the reviewer for the positive feedback and for recognizing our novel synergistic pipeline. Below we address each weakness.
>
> ---
> > **W1: Overselling System Acceleration (Hardware/System Level)**
> >
>
> The 7.0x end-to-end speedup vs. baseline (Nuggets) is driven by three compounding system-level savings from training only ~1% of total parameters (e.g., 71.9M/7.6B for Qwen-2.5-7B):
> - **Compute:** Weight-gradient General Matrix Multiply (GEMM) ($\partial L/\partial W$) for the frozen 99% is completely bypassed.
> - **Memory:** Adam optimizer states (momentum+variance) for 99% of parameters are never loaded or updated.
> - **Communication:** Frozen parameters are excluded from ZeRO-2's Reduce-Scatter payload, dismantling the cross-GPU synchronization wall, the dominant bottleneck in distributed training.
>
> **Table 11** disentangles both contributions via ablation:
>
> | Setting | AER | Speedup (vs. Full SFT) |
> |---|---|---|
> | Full SFT (100% data, 100% params) | 1.00 | 1× |
> | Param Sparsity only (100% data, 10% heads) | 0.58 | ~1.72× |
> | Data Sparsity only (10% data, 100% params) | 0.45 | ~2.22× |
>
> Both contributions are substantial and comparable, structural parameter pruning alone accounts for ~1.72× speedup. Together, they contribute to the 7.0× end-to-end speedup vs. Nuggets baseline, confirming the acceleration is not merely an illusion of data reduction.
>
> ---
> > **W2. Logical Gap in the Proxy Model**
> >
>
> The low proxy accuracy reflects a deliberate decoupling: *inference capability* requires thousands of steps and extensive data, but *structural sensitivity*, which heads route task-specific gradients, stabilizes within the first few steps (Sec. 5, FHI; Appendix D; [1-3]). Our proxy is a structural probe, not a solver. Mechanistic interpretability literature shows that task-sensitive routing circuits emerge early in gradient descent, well before task accuracy converges [1-3].
>
> To empirically validate this decoupling, we compare 20-step vs. 50-step proxies:
>
> | Method | GSM8K | DialogSum | BioInstruct |
> |---|---|---|---|
> | 20-step proxy | 18.45 | 12.30 | 10.15 |
> | P2D (20-step) | 78.82 | **44.04** | 32.74 |
> | 50-step proxy | 52.78 | 31.37 | 33.19 |
> | P2D (50-step) | **79.37** | 42.18 | **33.61** |
>
> Despite a dramatic proxy accuracy gain (18.45→52.78 on GSM8K), downstream P2D performance is nearly identical while linearly increasing overhead. Notably, P2D-DialogSum even regresses (44.04→42.18) with the 50-step proxy, confirming structural patterns are fully encoded within 20 steps. The structural importance distribution stabilizes very early—not a fragile artifact, but a highly precise signal for structural localization, ultimately driving our end-to-end SOTA results.
>
> ---
> > **W3. Baseline Fairness (Strawman Fallacy)**
> >
>
> We set $r=128$ for strict iso-parameter comparison (~1.0% trainable params, matching P2D) as in Appendix A.2. Following your suggestion, we conducted a rank search under the same 10% data constraint:
>
> | Method | Trainable Params | GSM8K | DialogSum | BioInstruct |
> |---|---|---|---|---|
> | LoRA (r=128) | ~1.0% | 75.82 | 42.86 | 32.08 |
> | LoRA (r=64) | ~0.6% | 77.19 | 40.38 | 31.13 |
> | LoRA (r=32) | ~0.3% | 76.31 | 38.23 | 30.27 |
> | **P2D (Ours)** | **~1.0%** | **78.82** | **44.04** | **32.74** |
>
> All LoRA variants consistently lag behind P2D. The iso-parameter design (Appendix A.2) confirms it is not a capacity artifact. The fundamental difference is **Synergy**: LoRA applies a uniform low-rank update across all layers, blind to the data being fed. P2D establishes a precise lock-and-key synergy between the curated 10% data and the exact structural heads it maximally activates, escaping the capacity-overfitting dilemma and confirming **Synergy**, not parameter count, is decisive.
>
> ---
> > **W4. Knowledge Acquisition Bottleneck (Frozen MLPs)**
> >
>
> Under extreme data constraints (10% budget), the key challenge is the **new knowledge storage vs. catastrophic forgetting** trade-off: aggressively updating heavy MLPs on a tiny dataset forces memorization of a narrow fact subset while overwriting the pre-trained knowledge base. Furthermore, recent research demonstrates that attention heads are the primary drivers of task-specific adaptation in alignment settings [2,3]. Table 1 empirically validates this on BioInstruct: *Full Parameters* (updates all MLPs) scores only 31.12 and LoRA scores 30.89 under 10% data, while $P2D^\ddagger$ (zero MLP updates) achieves 32.78 by routing the model's pre-existing knowledge through task-sensitive attention heads. In low-data regimes, sparse adaptation that protects the knowledge base is more robust than brute-force injection.
>
> ---
> **References**
>
> [1] Zhao, H, et al. "Explainability for large language models: A survey." ACM TIST, 2024.
>
> [2] Chen, H, et al. "Alps: Attention localization and pruning strategy for efficient alignment of large language models." ACL 2025.
>
> [3] Zhou, Z, et al. "On the role of attention heads in large language model safety." ICLR 2025.

---

> > ### Author Rebuttal · Reviewer_1zdm · 2026-04-04
> >
> > Thanks for your rebuttal. Overall, I think the rebuttal resolves most of my major concerns, although a few points are still only partially addressed. In particular, the response on baseline fairness is convincing, while the discussions on the claimed system-level speedup and the proxy model help clarify the method but do not completely remove my doubts. The concern about freezing MLP layers is also not fully resolved. On balance, I appreciate the contribution and the improved clarity after rebuttal, so I choose to retain my original score.

---

> > > ### Author Response · Authors · 2026-04-04
> > >
> > > We thank the reviewer for the continued discussion. We address the three remaining points with additional evidence below.
> > >
> > > ---
> > > **1. On system-level speedup: quantitative system profiling.**
> > >
> > > The speedup is a direct result of dismantling the **communication wall** in distributed training. P2D trains only ~1% of total parameters ($\rho_{train} \approx 1\%$, e.g., 71.9M/7.6B for Qwen-2.5-7B), yielding three compounding savings:
> > >
> > > - **Compute**: Weight-gradient GEMM ($\partial L / \partial W$) for the frozen 99% is entirely bypassed.
> > > - **Memory**: Adam optimizer states (momentum+variance) for 99% of parameters are never allocated or updated.
> > > - **Communication**: In ZeRO-2, gradient synchronization (Reduce-Scatter + All-Gather) operates over trainable parameters. For full SFT ($P_{train} \approx 7.6\text{B}$): $2 \times 7.6\text{B} \times 2\text{ bytes} \approx 30.4\text{ GB/step/GPU}$. For P2D: $0.01 \times 30.4\text{ GB} \approx 0.30\text{ GB/step/GPU}$, a 100× reduction that dismantles the dominant cross-GPU synchronization bottleneck.
> > >
> > > Table 11 confirms: parameter sparsity alone delivers 1.72× speedup, data sparsity 2.22×. This is not a data-reduction illusion. It is structural sparsity acting simultaneously on compute, memory, and communication. The savings are hardware-validated and compound with scale: Hardware-sensitivity experiments (detailed for Reviewer 94mv) confirm this quantitatively: AER drops 0.62→0.54 (4×A100→8×A100) and 0.54→0.48 (8×H200), as more GPUs and faster compute amplify Full SFT's 30.4 GB payload cost while P2D's 0.30 GB remains nearly unaffected. The advantage also compounds with scale: at 70B+, GQA shrinks $\rho_{train}$ further and larger GPU counts amplify ZeRO savings. As detailed for Reviewer 5QcZ (Q1, W2, and cross-scale timing), 14B/32B results confirm P2D$^\ddagger$ surpasses Full SFT, and $W_1$ timing (\~30s→100s) vs. gradient scoring (\~10min→5.7h) shows the efficiency gap widens with scale.
> > >
> > > **2. On the proxy model: structural circuits stabilize before task convergence.**
> > >
> > > We acknowledge that the reliability of early-stage proxies is a legitimate concern, and we offer the following evidence that our 20-step proxy is not brittle. Our ablation shows downstream performance is nearly identical (78.82 vs. 79.37 on GSM8K) despite a dramatic accuracy jump (18.45→52.78). This is not surprising in light of recent mechanistic interpretability work. [1] tracked circuits across 300B training tokens in decoder-only LLMs (70M–2.8B) and found that **task-supporting attention components emerge at similar token counts across model scales**, with the implemented algorithm remaining stable throughout training. Anthropic [2] shows that **task-sensitive routing structures emerge early in gradient descent as stable, fundamental building blocks of transformer computation**. FHI probes this early-emerging structure, not the final task-solving capability, which explains why 20 steps suffice and more steps do not help.
> > >
> > > **3. On frozen MLPs: alignment re-routes pre-existing knowledge.**
> > >
> > > We acknowledge which modules to update remains an open question in the field. The community consensus over recent years
> > > has shifted substantially: recent work shows **for downstream alignment, attention modules are the primary drivers of task-specific adaptation**, while MLP-centric fine-tuning has received comparatively less focus.
> > >
> > > [3] show MLP layers store factual knowledge while attention heads govern retrieval. For GSM8K and BioInstruct, Qwen-2.5 is pretrained on extensive mathematical and biomedical corpora, so the relevant knowledge already exists. [4] shows fine-tuning primarily teaches routing rather than knowledge injection. **What alignment teaches is which knowledge to route, not what new facts to store.** Updating MLPs on a 10% data budget risks narrow memorization while overwriting the pretrained knowledge base. [5-8] all find that freezing MLPs while updating only sparse attention heads matches or exceeds LoRA, confirming attention-centric adaptation as the primary driver of alignment. Table 1 provides direct empirical confirmation: **$P2D^\ddagger$ achieves 32.78 with zero MLP updates**, outperforming Full SFT (31.12) and LoRA (30.89) on BioInstruct. We will discuss this in the Limitations section of the final version.
> > >
> > > ---
> > > **References**
> > >
> > > [1] LLM Circuit Analyses Are Consistent Across Training and Scale. NeurIPS 2024.
> > >
> > > [2] A Mathematical Framework for Transformer Circuits. Anthropic, 2021.
> > >
> > > [3] Transformer Feed-Forward Layers Are Key-Value Memories. EMNLP 2021.
> > >
> > > [4] LIMA: Less Is More for Alignment. NeurIPS 2023.
> > >
> > > [5] HeadMap: Locating and Enhancing Knowledge Circuits in LLMs. ICLR 2025.
> > >
> > > [6] LoFiT: Localized Fine-tuning on LLM Representations. NeurIPS 2024.
> > >
> > > [7] Alps: Attention Localization and Pruning Strategy. ACL 2025.
> > >
> > > [8] On the Role of Attention Heads in LLM Safety. ICLR 2025.

---

### Official Review · Reviewer_94mv · 2026-03-11

**Soundness:** 2
**Presentation:** 3
**Significance:** 3
**Originality:** 2
**Overall Recommendation:** 3
**Confidence:** 4

**Summary:**

The paper introduces P2D, a unified pipeline for efficient LLM alignment that posits the "Strong Map Hypothesis”. It says a sparse subset of attention heads dominates task-specific adaptation. P2D uses a lightweight proxy for fast head identification, employs these heads to guide ICL-based data selection, and sparsely fine-tunes only those heads on the curated 10% subset, achieving 8.3 pp gains and 7.0 times speedup over baselines like LoRA, LoFiT, IFD, Nuggets, and Data Whisperer on 3 datasets (GSM8K, DialogSum, and BioInstruct) with Qwen/Llama models (Table 1).

**Compliance With Llm Reviewing Policy:**

Affirmed.

**Key Questions For Authors:**

1. Eq. 6 uses W1 on softmax-flattened composite matrices. Why not direct activation gradients or Fisher info (as hinted Table 10)? Ablation vs alternatives could strengthen FHI claims.
2. Strong Map Hypothesis claims sparse heads. Did you verify by fine-tuning random heads on P2D data (or vice versa) to isolate synergy?
3. AER includes proxy training but assumes fixed hardware. How sensitive is it to other accelerators or fewer in number?
4. Domain tasks (DialogSum/BioInstruct) lag full SFT more than GSM8K (Table 1). Is P2D less effective for low-data domains?
5. What strategy was used for choosing (x_i, y_i) pairs in eq 4? How did you batched original data?
6. In sec 3.2.1, did you just sampled 100 examples once and made the claims on that basis or repeated sampling few times?

**Limitations:**

No any

**Strengths And Weaknesses:**

* Soundness: Claims are supported by solid experiments across 3 models/datasets.
* Presentation: Well-structured narrative. Line 185, one closing bracket is missing, left side
* Significance: Addresses key efficiency gap in LLM adaptation (data+PEFT coupling), AER enables holistic eval, 7 times speedup practical for domain adaptation; impact specialized to instruction-tuning but broadens via synergy insight.
* Originality: Paper address significant issue in current LLM practice but overall originality is moderate. It presents unified pipeline repurposing heads for data selection (vs isolated baselines), high-affinity validation (Fig. 4).

---

> ### Author Rebuttal · Authors · 2026-03-31
>
> We sincerely thank the reviewer for the thorough evaluation and for recognizing the significance of addressing the data+PEFT coupling gap, with the holistic evaluation enabled by AER. The bracket typo on Line 185 is now fixed.
>
> ---
> > **Q1. Eq. 6 uses W1. Why not direct activation gradients or Fisher info?**
>
> We have corrected Appendix. D typos to reflect our actual $W_1$ implementation. Our choice of static $W_1$ is driven by two key advantages:
>
> **1. Mathematical Superiority for Small Shifts:** The $W_1$ distance between $P^h_B$ (base) and $P^h_T$ (proxy, softmax-flattened weight matrices), under a small parameter translation ($P_T^h\approx P_B^h(x-\Delta)$), scales **linearly**: $W_1(P_B^h,P_T^h)\approx\\|\Delta\\|$. Conversely, gradient-based scoring $D_{GB}$ requires calibration inputs and is defined as:
> $$D_{GB}(\pi_B^h(x)\\|\pi_T^h(x))=\sum_i \pi_B^h(x_i)\log\frac{\pi_B^h(x_i)}{\pi_T^h(x_i)}$$
> where $\pi_B^h(\cdot|x)$ is head $h$'s activation distribution given calibration input $x$. Treating $\pi^h(x;\theta)$ as parametric, the second-order Taylor expansion around $\theta_B$ gives $D_{GB}\approx\frac{1}{2}\mathcal{I}(\pi_B^h)\\|\Delta\\|^2$ (**quadratic**). When $\\|\Delta\\|\ll 1$, this quadratic suppression disproportionately buries critical structural signals into noise, unlike $W_1$'s linear and robust sensitivity.
>
> **2. Computational Efficiency:** $W_1$ performs data-free static weight comparisons with near-zero overhead, unlike Activation Gradients that require expensive forward/backward passes. The ablation confirms $W_1$ matches dynamic performance while reducing scoring time:
>
> | Metric | GSM8K | DialogSum | BioInstruct | Scoring Time |
> |---|---|---|---|---|
> | cosine similarity | 63.28 | 28.37 | 21.43 | ~10s |
> | gradient-based | 76.48 | 43.79 | **33.08** | ~10min |
> | $W_1$ (Ours) | **78.82** | **44.04** | 32.74 | ~30s |
>
> ---
> > **Q2. Did you verify by fine-tuning random heads on P2D data (or vice versa) to isolate synergy?**
>
> We conducted a strict cross-ablation to isolate synergy from both directions:
>
> | ρD | ρP | GSM8K | DialogSum | BioInstruct |
> |---|---|---|---|---|
> | P2D Data | Random Heads | 67.13 | 38.77 | 25.74 |
> | Random Data | P2D Heads | 72.32 | 40.33 | 27.98 |
> | IFD/Nuggets/DW | P2D Heads | 71.42/68.91/75.84 | 36.06/35.07/42.12 | 23.82/25.91/30.19 |
> | **P2D Data** | **P2D Heads** | **78.82** | **44.04** | **32.74** |
>
> This confirms optimal efficiency requires precisely paired data and parameters.
>
> ---
> > **Q3. How sensitive is AER to other accelerators or fewer GPUs?**
>
> **1. FLOPs Lower Bound:** Note: for simplicity, AER calculation here covers training only. The compute costs for forward ($C_{fwd}$), backward activation gradients ($C_{ba}$), and backward weight gradients ($C_{bw}$) are approximately equal. Crucially, P2D updates only $\rho_P=10\%$ of attention heads, whose parameters account for a small fraction of total parameters, yielding a tiny actual trainable ratio $\rho_{train}$ ($0.94\%$ for Q2.5-7, $1.47\%$ for Q3-8, $1.34\%$ for L3-8). The weight gradient cost is thus nearly bypassed:
> $$\text{AER}\_{bound}\approx\frac{C\_{fwd}+C\_{ba}+\rho\_{train}\cdot C\_{bw}}{C\_{fwd}+C\_{ba}+C\_{bw}}=\frac{2+\rho\_{train}}{3}\approx\frac{2.01}{3}\approx 0.67$$
>
> **2. Empirical Scaling:** Wall-clock AER is often lower due to eliminated optimizer state I/O and ZeRO-2 communication. A controlled evaluation (Batch=128) confirms:
>
> | Hardware | AER |
> |---|---|
> | 8×A100 | 0.54 |
> | 4×A100 | 0.62 |
> | 8×H200 | 0.48 |
> | 4×H200 | 0.55 |
>
> More GPUs amplify ZeRO-2 savings; H200's higher compute density further lowers AER.
>
> ---
> > **Q4. Is P2D less effective for low-data domains?**
>
> The gap reflects task dynamics (Lines 318–322): domain tasks require new knowledge unlike GSM8K's pattern activation, so data reduction causes larger drops. Under strictly constrained data budget, P2D is more suitable for these scenarios, which selects information-dense examples while targeted head updates preserve pre-trained knowledge and prevent overfitting.
>
> ---
> > **Q5. What strategy was used for (x_i, y_i) pairs in Eq. 4? How was data batched?**
>
> Eq. 4 is a theoretical Leave-One-Out formulation, we approximate it via ICL-Based Probing (Eq. 7–10). In each iteration, we randomly sample **5 examples** as demonstrations ($\mathcal{D}\_d$) and evaluate **3 candidate queries** without replacement. Crucially, randomly resampling $\mathcal{D}\_d$ each iteration serves as a **bias-reduction mechanism**: each candidate is evaluated against diverse demonstration contexts, making $s\_{final}$ robust to any demonstration choice and reflecting intrinsic task relevance (Appendix F).
>
> ---
> > **Q6. Did you sample 100 examples once or repeat sampling?**
>
> All experiments use **three distinct random seeds** (42, 43, 44), per Appendix A.2 (Line 580). The global seed controls the 100-sample proxy subset, so results average across three independent subsets. Consistent SOTA performance verifies FHI's robustness against sampling artifacts.

---

> > ### Author Rebuttal · Reviewer_94mv · 2026-04-02
> >
> > The rebuttal provides clear and detailed responses to most technical questions, particularly for Eq. 6 and the sampling strategies. The random vs. P2D head/data ablations also directly address the Strong Map Hypothesis concern. The distinctions drawn between high-data and low-data domain performance (GSM8K vs. DialogSum/BioInstruct) are reasonable and well contextualized. However, the discussion on AER’s hardware sensitivity remains somewhat qualitative. Overall, the rebuttal strengthens the paper’s soundness and clarity, but as the core contribution remains largely orchestration rather than a fundamentally novel idea, my evaluation and score remain unchanged.

---

> > > ### Author Response · Authors · 2026-04-03
> > >
> > > We thank the reviewer for the continued engagement and address both remaining points.
> > >
> > > ---
> > > **1. P2D is not merely orchestration.**
> > > Our core contribution reveals a new inductive bias of LLMs: task-specific data maps to a sparse, consistent set of task-specific parameters. **If P2D were simply combining data selection and PEFT, replacing either component with a SOTA alternative should yield comparable results. Our cross-ablation proves otherwise.** Pairing IFD, Nuggets, or Data Whisperer with P2D heads all score substantially lower (GSM8K: 71.42/68.91/75.84 vs. 78.82), and pairing P2D data with random heads collapses to 67.13. **Only the precise data-head pairing achieves peak performance, proving Lock-and-Key Synergy is the mechanism, not an artifact.** P2D's four distinct contributions are:
> > >
> > > - **Strong Map Hypothesis (Empirical Innovation).** No prior work has identified that a sparse, consistent subset of attention heads dominates task-specific adaptation across models and tasks, nor demonstrated cross-task hub heads. This is a foundational finding about LLM structure.
> > > - **Lock-and-Key Synergy (Design Innovation).** Prior data selection and PEFT methods were studied independently. P2D exposes a bidirectional coupling: task-sensitive heads (identified via $W\_1$) guide data selection by scoring candidates through their activations, ensuring the selected data specifically captures the knowledge patterns these heads encode. Conversely, the selected data trains exclusively those same heads, ensuring parameter updates are targeted at exactly the heads that are most responsive to the chosen examples. Neither direction alone suffices, as the cross-ablation confirms: the selected data with random heads, or P2D heads with generic data, both collapse. Only the mutually-informed pairing unlocks the full synergy.
> > > - **$W\_1$ Metric (Theoretical Innovation).** We formally show $W\_1 \approx \|\Delta\|$ scales linearly, while Fisher/gradient methods scale quadratically ($D_{GB} \approx \frac{1}{2}\mathcal{I}\|\Delta\|^2$), disproportionately suppressing small but critical signals when $\|\Delta\| \ll 1$ after only 20 proxy steps. This is a new data-free theoretical justification for static weight-space head identification.
> > > - **AER (Evaluation Innovation).** A unified metric enabling rigorous efficiency comparison across methods with different data, parameter, and compute trade-offs.
> > >
> > > **2. On AER hardware sensitivity: strict quantitative profiling.**
> > > In ZeRO-2, each step involves two communication collectives: (1) **Reduce-Scatter** to synchronize gradients ($\approx P_{train} \times 2$ bytes per GPU), and (2) **All-Gather** to broadcast updated parameters ($\approx P_{train} \times 2$ bytes per GPU). Total per-GPU communication per step: $\approx 2P_{train} \times 2$ bytes (bf16).
> > >
> > > For a 7B model under full SFT ($P_{train} \approx 7.6\text{B}$):
> > > $$\text{ZeRO-2 traffic} = 2 \times 7.6\text{B} \times 2\text{ bytes} \approx 30.4\text{ GB/step/GPU}$$
> > >
> > > P2D freezes 99% of parameters ($\rho_{train} \approx 1\%$), so only trainable heads participate in gradient communication:
> > > $$\text{P2D traffic} = 0.01 \times 30.4\text{ GB} \approx 0.30\text{ GB/step/GPU}$$
> > >
> > > This is a strict in ZeRO-2 traffic. Additionally, ZeRO-2 partitions optimizer states across GPUs. Frozen parameters carry no optimizer state, eliminating a further ~99% of optimizer I/O.
> > >
> > > **GPU count scaling (4×A100 → 8×A100, AER: 0.62 → 0.54).** In a ring topology, per-GPU communication volume scales as $\frac{N-1}{N}$: at $N=4$ this is $0.75\times$, at $N=8$ it is $0.875\times$, a modest 16.7% volume increase. However, doubling GPU count simultaneously exacerbates fabric contention and cross-node synchronization latency. This network saturation severely impacts Full SFT's 30.4 GB payload at every synchronization step, whereas P2D's 0.30 GB payload is virtually immune to congestion at any GPU count, widening the absolute savings and pushing AER lower.
> > >
> > > **Hardware density scaling (8×A100 → 8×H200, AER: 0.54 → 0.48).** H200 delivers 3.17× higher BF16 throughput (989 vs. 312 TFLOPS) but only 1.5× NVLink bandwidth (900 vs. 600 GB/s). As compute accelerates by 3.17× while bandwidth grows only 1.5×, the 30.4 GB ZeRO-2 payload constitutes a proportionally larger fraction of total step time on H200. P2D's reduction in this traffic is therefore relatively more impactful on H200: it is eliminating the dominant communication bottleneck (not the already-fast compute) that drives AER from 0.54 to 0.48. We will include this profiling in the revised appendix.
> > >
> > > ---
> > > We hope this quantitative profiling and the cross-ablation evidence clarify P2D's fundamental contributions. As the discussion period continues, please let us know if you have any follow-up questions or lingering concerns. We remain entirely at your disposal and would be more than happy to elaborate further.

---

### Decision · Program_Chairs · 2026-04-30

**Decision:**

Accept (regular)

**Comment:**

This paper proposes P2D for adapting LLMs to specialized domains efficiently. The method first identifies critical attention heads via a lightweight proxy model that is obtained by fine-tuning the base model on a small subsampled subset of data for small number of steps. This proxy model is used for both sample mining and structural pruning. The paper also introduces the AER metric that normalizes the total alignment cost against full fine-tuning. The paper claims updating 10% of heads on 10% of data achieves strong performance with 7x end-to-end speedup.

Reviewers recognized the novel coupling of data selection and parameter-efficient fine-tuning, the useful AER efficiency metric, and strong empirical analysis, but raised concerns about the proxy model's reliability, the novelty of the pipeline, system-level speedup validity, and frozen-MLP limitations. During the rebuttal, the authors expanded empirical results to additional scales (Qwen2.5-1.5B-Instruct, 3B, and 32B), provided cross-scale evidence of the advantage of Wasserstein-1 distance for computing the task sensitivity score, added analysis of communication savings including in ZeRO-2 optimizer gradient synchronization, and demonstrated results on more general benchmarks (MMLU and ARC-C). After the rebuttal, Reviewer edZA raised their score to Weak Accept, two maintained Weak Accept (Reviewers 1zdm and 5QcZ), while the Weak Reject reviewer (94mv) remained unconvinced on novelty.

In discussions with AC, Reviewer 94mv remained concerned that the paper does not provide a controlled ablation comparing their method with other parallel efficiency techniques under the same hardware conditions and missing study of alternatives to ZeRO-2 distributed training strategy that makes it difficult to isolate the contribution of the proposed approach relative to existing efficiency methods. At the same time Reviewer 1zdm found the authors' response about system-level speedup and the breakdown of end-to-end pipeline efficiency convincing. Reviewer 1zdm remained concerned that the frozen-MLP discussion does not fully rule out a knowledge-acquisition bottleneck beyond the evaluated tasks. The AC acknowledges these concerns while recognizing that the existing results including additions in the rebuttal are useful contributions to the community that could be improved in the future and extended to more optimizers and hardware conditions. The AC recommends acceptance contingent on incorporating the scaling experiments and addressing the interpretive limitations (proxy model stability, frozen MLP) in the revision.